# Hamiltonian Asymmetric Fusion: One-Way Safe Directed Refinement under Modality Imbalance

**Bingbing Chen** [* 1 2]  **Congcong Liu** [* 2]  **Dong Liang** [2 3]  **Zhuo-Xu Cui** [2 3]

## Abstract

In RGB–D salient object detection, multimodal fusion is commonly implemented via symmetric token interaction, implicitly allowing information to flow in both directions. Under *modality imbalance*—when an auxiliary stream is substantially noisier than a designated primary stream—such symmetry creates a *backflow channel* that injects auxiliary noise into the primary representation and amplifies errors across iterative refinement stages. We formulate fusion in this regime as *directed refinement with one-way safety*: the primary modality defines a guidance field, while only auxiliary representations are iteratively purified, and primary perturbations induced by the auxiliary stream are explicitly bounded. We propose *Hamiltonian Asymmetric Fusion* (HAF), a lightweight unrolled refinement block that updates auxiliary tokens with momentum regularization and gated driving. The refinement force is instantiated by FFT-based spectral global correlation and modulated by a shared learnable spectral response to emphasize reliable frequency components with minimal parameters; a leaky momentum gate and a stable integrator improve multi-step refinement stability. We provide guarantees of auxiliary error contraction and bounded primary perturbation, which symmetric fusion operators do not satisfy under imbalance. Experiments on six RGB–D SOD benchmarks show consistent gains and substantially more graceful degradation under controlled auxiliary corruption.

## 1. Introduction

Multimodal learning improves robustness by integrating complementary cues across heterogeneous sensors (Baltrušaitis et al., 2018). In RGB–D salient object detection (SOD), depth provides geometric structure that helps delineate object boundaries and resolve textureless regions, while RGB supplies appearance cues for semantic discrimination (Zhou et al., 2021a; Chen et al., 2025). Modern RGB–D SOD systems typically rely on progressive decoding and stage-wise refinement, where cross-modal interaction is repeatedly applied across feature hierarchies (Sun et al., 2023; Hu et al., 2024; Luo et al., 2024).

**Modality imbalance induces backflow under iterative fusion.** While prior work often emphasizes modality gaps and alignment, we focus on a different failure mode that becomes critical under repeated refinement: *modality imbalance*, where an auxiliary stream is substantially noisier or less reliable than a designated primary stream. In this regime, *symmetric* fusion (e.g., bidirectional cross-attention or symmetric diffusion-style coupling) creates an unavoidable *backflow channel*: auxiliary noise is injected into the primary representation and can accumulate across refinement steps, leading to unstable refinement trajectories and degraded dense predictions (Wang & Yu, 2025). Thus, the central problem is to design fusion operators that remain stable across multi-step refinement in the presence of highly asymmetric modality reliability.

**Directed refinement with one-way safety.** We argue that fusion under modality imbalance should be formulated as a *directed refinement process* with an explicit safety requirement. The primary modality should define a guidance field for refinement, while the auxiliary representation is iteratively purified; meanwhile, the primary stream should be protected from auxiliary-driven perturbations. We refer to this principle as *one-way safety*: auxiliary errors contract through refinement and primary perturbations remain bounded. This contrasts with symmetric interaction operators—including cross-attention and Cross-Diffusion Attention (CDA) (Wang et al., 2024)—which structurally permit backflow and do not explicitly encode role-asymmetric influence across multiple refinement steps. Fig. 1 provides a schematic comparison of symmetric inter-

---
[*]Equal contribution [1]School of Biomedical Engineering, ShanghaiTech University, Shanghai 201210, China [2]Shenzhen Institutes of Advanced Technology, Chinese Academy of Sciences, Shenzhen, China [3]Guangdong Provincial Key Laboratory of Multimodality Non-Invasive Brain-Computer Interfaces. Correspondence to: Zhuo-Xu Cui <zx.cui@siat.ac.cn>, Dong Liang <dong.liang@siat.ac.cn>.

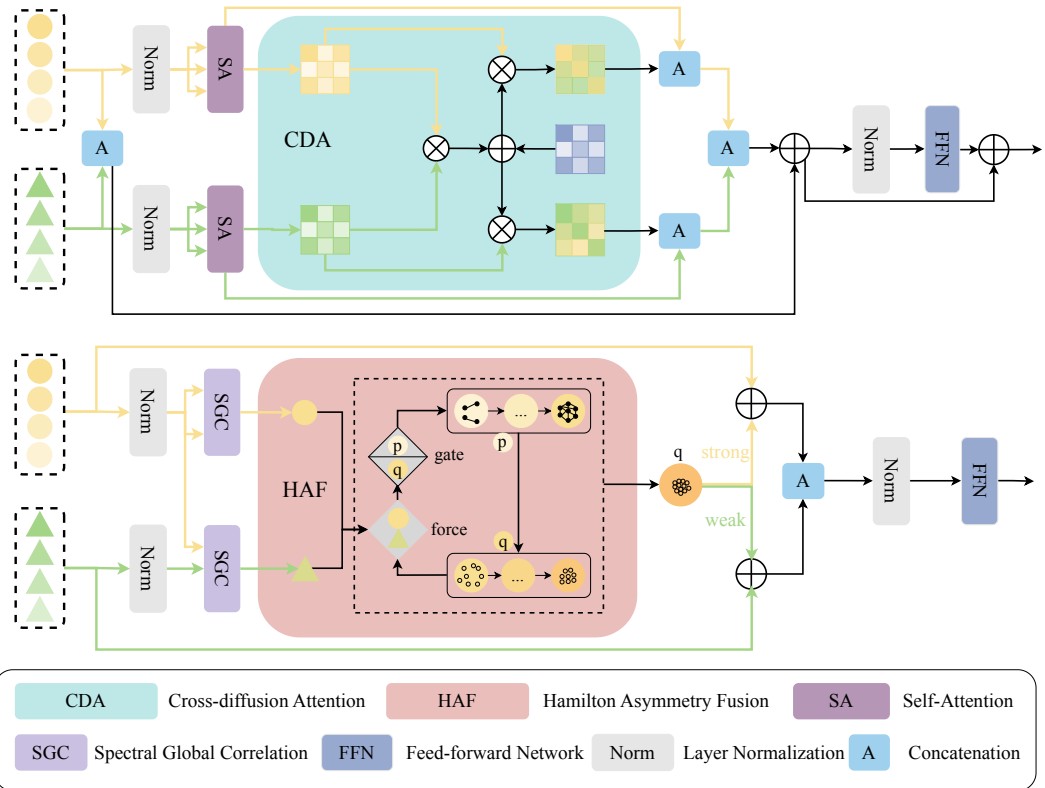

*Figure 1.* Comparison between CDA and HAF. **Top:** CDA performs *symmetric* cross-diffusion by mutually propagating information between modality-specific attention maps. **Bottom:** HAF formulates the fusion process as an unrolled Hamiltonian evolution with *asymmetric* residual connections.

action versus role-asymmetric refinement.

**Hamiltonian Asymmetric Fusion (HAF).** Based on this principle, we propose **Hamiltonian Asymmetric Fusion (HAF)**, a lightweight fusion block that implements directed refinement via an unrolled, momentum-regularized update on auxiliary tokens driven by a primary-induced potential. HAF combines (i) a global correlation-based driving force with spectral adaptation and (ii) gated momentum and a stable integrator to regulate multi-step refinement. Importantly, HAF is *role-asymmetric*: the primary stream acts as an anchor that induces refinement, while only the auxiliary state is updated; the refined auxiliary information is then injected back with asymmetric residual weights to reduce cross-modal interference. In our RGB–D SOD setting, we use a task-defined role assignment (depth as primary, RGB as auxiliary), while the formulation can be applied under alternative role assignments.

**Results.** We evaluate HAF on six RGB–D SOD benchmarks and observe consistent improvements in $S$-measure, max $F_\beta$, max $E_\xi$, and MAE over strong Transformer-based baselines. We further conduct unified module-swapping and controlled auxiliary-corruption experiments to isolate fusion behavior under modality imbalance, where HAF demonstrates graceful degradation than competing fusion plugins.

**Contributions.**

- We identify *modality imbalance* under iterative refinement as a key failure mode and propose *directed refinement with one-way safety* to prevent auxiliary-to-primary backflow contamination.

- We introduce *HAF*, a role-asymmetric fusion block that updates only auxiliary tokens through unrolled momentum-regularized refinement driven by the primary modality.

- We validate consistent gains and improved robustness via unified module-swapping and controlled auxiliary corruption across six benchmarks.

**Conflict of Interest Disclosure.** The authors declare no financial conflicts of interest related to this work.

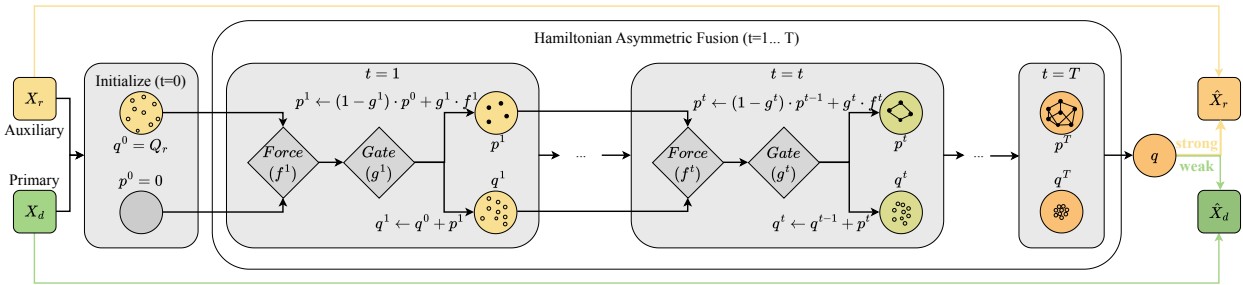

*Figure 2.* Overall pipeline of HAF. Starting from $\mathbf{q}^0 = \mathbf{Q}_r$ and $\mathbf{p}^0 = \mathbf{0}$, HAF unrolls $T$ steps where a force term is computed via spectral global correlation and modulated by a learned gate to update $(\mathbf{p}^t, \mathbf{q}^t)$. The final state $\mathbf{q}^T$ is injected back to both modalities with asymmetric residual weights.

## 2. Method

We study feature fusion under *unequal reliability* between two modalities, e.g., RGB as a noisy *auxiliary* stream and depth as a cleaner *primary* stream. Let $\mathbf{X}_r, \mathbf{X}_d \in \mathbb{R}^{n \times d}$ denote token maps flattened over space, where $n = HW$ and $d$ is the feature dimension. Most existing fusion blocks rely on *symmetric* bidirectional interactions (e.g., cross-attention), implicitly assuming comparable signal-to-noise ratio (SNR) across modalities. Under *modality imbalance*, where $\mathbf{X}_r$ is substantially noisier, symmetric coupling inevitably trades auxiliary utility against *noise backflow* into the cleaner primary stream $\mathbf{X}_d$, degrading primary representations.

We introduce *Hamiltonian Asymmetric Fusion* (HAF), a directed interaction mechanism that structurally prevents such backflow. HAF treats the primary stream as a fixed *guiding potential field* and refines only the auxiliary tokens by evolving them within this field. Crucially, the primary stream is not updated as a function of auxiliary features (except for an optional, explicitly controlled residual scaled by a small learnable scalar), which enforces one-way safety at the block level. Fig. 2 presents the overall architecture.

### 2.1. Spectral Global Correlation

To capture long-range structure without instantiating a dense $n \times n$ affinity matrix, the design follows GFNet's Fourier-domain global mixing paradigm (Rao et al., 2021). The input-independent global filter is instantiated as a query–key spectral correlation operator (denoted as *SGC* in this work), and a *shared learnable spectral response* is incorporated to reweight frequency components, prioritizing reliable bands under auxiliary corruption.

Given tokens $\mathbf{X} \in \mathbb{R}^{n \times d}$, we form query/key projections $\mathbf{Q}, \mathbf{K} \in \mathbb{R}^{n \times d}$ via linear layers, and reshape them channel-wise (and per head) into $H \times W$ feature maps. Let $\mathcal{F}$ and $\mathcal{F}^{-1}$ denote the 2D FFT and its inverse, applied independently to each channel (and head). We apply a Hann window $\mathbf{w} \in \mathbb{R}^{H \times W}$ to reduce boundary artifacts and spectral leak-

age. We further introduce a learnable frequency response $\mathbf{G} \in \mathbb{R}^{H \times W}$ that is *shared across channels and heads*, adding only $O(HW)$ parameters while allowing the model to reweight frequency bands based on reliability.

We define the (augmented) SGC operator as

$$\mathrm{SGC}(\mathbf{Q}, \mathbf{K}) = \mathcal{F}^{-1}\Big(\mathbf{G} \odot \big(\mathcal{F}(\mathbf{Q} \odot \mathbf{w}) \odot \overline{\mathcal{F}(\mathbf{K} \odot \mathbf{w})}\big)\Big), \tag{1}$$

where $\odot$ denotes element-wise multiplication and $\overline{(\cdot)}$ is complex conjugation. Eq. (1) computes dense correlation fields in $O(n \log n)$ time and $O(n)$ memory, avoiding explicit construction of $n \times n$ attention weights.

We use SGC both intra-modally and cross-modally. Within-modality correlations are $\mathbf{U}_r = \mathrm{SGC}(\mathbf{Q}_r, \mathbf{K}_r)$ and $\mathbf{U}_d = \mathrm{SGC}(\mathbf{Q}_d, \mathbf{K}_d)$. For directed cross-modal interaction, we compute depth-to-RGB correlation

$$\mathbf{U}_{r \leftarrow d} = \mathrm{SGC}(\mathbf{Q}_r, \mathbf{K}_d), \tag{2}$$

which correlates auxiliary queries with primary keys and will be used to instantiate the primary-induced force field for one-way refinement.

### 2.2. Hamiltonian Asymmetric Fusion

We reinterpret fusion as a controlled refinement process in which *only* the auxiliary tokens evolve. Let $\mathbf{q} \in \mathbb{R}^{n \times d}$ denote the auxiliary state to be refined and $\mathbf{p} \in \mathbb{R}^{n \times d}$ its momentum. We define a Hamiltonian

$$H(\mathbf{q}, \mathbf{p}) = \tfrac{1}{2}\|\mathbf{p}\|^2 + V(\mathbf{q}), \tag{3}$$

where $V(\mathbf{q})$ is a potential energy induced by correlations with the primary modality. The corresponding continuous-time Hamiltonian dynamics are

$$\dot{\mathbf{q}} = \frac{\partial H}{\partial \mathbf{p}} = \mathbf{p}, \qquad \dot{\mathbf{p}} = -\frac{\partial H}{\partial \mathbf{q}} = -\nabla_{\mathbf{q}} V(\mathbf{q}). \tag{4}$$

**Spectral force field.** We instantiate the force using SGC. Let $\mathbf{K}_d$ and $\mathbf{K}_r$ denote key projections from the primary

and auxiliary streams. We define

$$F_{\text{main}}(\mathbf{q}) = \text{SGC}(\mathbf{q}, \mathbf{K}_d), \qquad (5)$$

$$F_{\text{aux}}(\mathbf{q}) = \text{SGC}(\mathbf{q}, \mathbf{K}_r), \qquad (6)$$

and approximate the negative potential gradient as

$$-\nabla_{\mathbf{q}} V(\mathbf{q}) \approx F_{\text{main}}(\mathbf{q}) - \mathbf{q} + \gamma\big(\mathbf{q} - F_{\text{aux}}(\mathbf{q})\big), \quad (7)$$

where $\gamma \geq 0$ balances (i) attraction toward primary-supported patterns through $F_{\text{main}}(\mathbf{q}) - \mathbf{q}$ and (ii) repulsion from auxiliary-only patterns through $\mathbf{q} - F_{\text{aux}}(\mathbf{q})$. Intuitively, the second term suppresses structures that are self-consistent within the auxiliary stream but weakly supported by the primary.

**Symplectic refinement with *leaky* momentum gating.** We discretize the dynamics with a symplectic-style update, while introducing a learned gate that stabilizes multi-step refinement. We initialize $\mathbf{q}^0 = \mathbf{Q}_r$ and $\mathbf{p}^0 = \mathbf{0}$. For $t = 1, \ldots, T$, define the force

$$\mathbf{f}^t = F_{\text{main}}(\mathbf{q}^{t-1}) - \mathbf{q}^{t-1} + \gamma\big(\mathbf{q}^{t-1} - F_{\text{aux}}(\mathbf{q}^{t-1})\big), \quad (8)$$

and compute a gate

$$\mathbf{g}^t = \sigma\big(\text{MLP}\big(\mathbf{q}^{t-1} \,\|\, \mathbf{p}^{t-1}\big)\big), \qquad \mathbf{g}^t \in (0,1)^{n \times d}, \quad (9)$$

where $\|$ denotes concatenation and $\sigma(\cdot)$ is the sigmoid.

Crucially, we adopt a *leaky* (gated) momentum update

$$\mathbf{p}^t = (\mathbf{1} - \mathbf{g}^t) \odot \mathbf{p}^{t-1} + \mathbf{g}^t \odot \mathbf{f}^t, \qquad (10)$$

$$\mathbf{q}^t = \mathbf{q}^{t-1} + \mathbf{p}^t. \qquad (11)$$

Compared to the pure accumulation $\mathbf{p}^t = \mathbf{p}^{t-1} + \mathbf{g}^t \odot \mathbf{f}^t$, the leaky form in (10) explicitly damps stale momentum and is consistent with the stability assumptions used in our theory (Appendix C.4). Empirically, it improves robustness when the auxiliary stream is heavily corrupted, while preserving the benefits of multi-step refinement for small $T$ (typically $T \in \{2, 3\}$).

**Asymmetric residual fusion.** After $T$ refinement steps, we obtain $\mathbf{q}^T$ as a purified auxiliary state. We then output

$$\hat{\mathbf{X}}_r = \mathbf{X}_r + \alpha_{\text{aux}}\, \mathbf{q}^T, \qquad (12)$$

$$\hat{\mathbf{X}}_d = \mathbf{X}_d + \alpha_{\text{main}}\, \mathbf{q}^T, \qquad (13)$$

where $\alpha_{\text{aux}}, \alpha_{\text{main}} \in \mathbb{R}$ are learnable scalars. Setting $\alpha_{\text{main}} \approx 0$ enforces strict one-way fusion (no auxiliary-to-primary update), while a small non-zero $\alpha_{\text{main}}$ permits a controlled injection of refined auxiliary information into the primary stream.

**Algorithm.** Algorithm 1 summarizes one HAF block.

---

**Algorithm 1** Hamiltonian Asymmetric Fusion (HAF) Block

**Input:** auxiliary features $\mathbf{X}_r \in \mathbb{R}^{n \times d}$, primary features $\mathbf{X}_d \in \mathbb{R}^{n \times d}$, steps $T$, repulsion weight $\gamma$, fusion weights $\alpha_{\text{aux}}, \alpha_{\text{main}}$
**Output:** fused features $\hat{\mathbf{X}}_r, \hat{\mathbf{X}}_d$
// **Projection and initialization**
$\mathbf{Q}_r, \mathbf{K}_r \leftarrow \text{LinearProj}(\mathbf{X}_r)$
$\mathbf{Q}_d, \mathbf{K}_d \leftarrow \text{LinearProj}(\mathbf{X}_d)$
$\mathbf{q}^0 \leftarrow \mathbf{Q}_r$
$\mathbf{p}^0 \leftarrow \mathbf{0}$
**for** $t = 1$ **to** $T$ **do**
$\quad F_{\text{main}} \leftarrow \text{SGC}(\mathbf{q}^{t-1}, \mathbf{K}_d)$
$\quad F_{\text{aux}} \leftarrow \text{SGC}(\mathbf{q}^{t-1}, \mathbf{K}_r)$
$\quad \mathbf{f}^t \leftarrow F_{\text{main}} - \mathbf{q}^{t-1} + \gamma\big(\mathbf{q}^{t-1} - F_{\text{aux}}\big)$
$\quad \mathbf{g}^t \leftarrow \sigma\big(\text{MLP}(\mathbf{q}^{t-1} \,\|\, \mathbf{p}^{t-1})\big)$
$\quad \mathbf{p}^t \leftarrow (\mathbf{1} - \mathbf{g}^t) \odot \mathbf{p}^{t-1} + \mathbf{g}^t \odot \mathbf{f}^t$
$\quad \mathbf{q}^t \leftarrow \mathbf{q}^{t-1} + \mathbf{p}^t$
**end for**
$\hat{\mathbf{X}}_r \leftarrow \mathbf{X}_r + \alpha_{\text{aux}}\, \mathbf{q}^T$
$\hat{\mathbf{X}}_d \leftarrow \mathbf{X}_d + \alpha_{\text{main}}\, \mathbf{q}^T$

---

## 3. Theory: Stability and Robust Noise Contraction of the HAF Dynamics

We analyze the discrete-time dynamics implemented by the HAF block in Section 2, i.e., the coupled evolution of auxiliary states $(\mathbf{q}^t, \mathbf{p}^t)$ driven by spectral forces instantiated by SGC and *leaky (gated) momentum*. Our goals are to establish: (i) existence of an invariant (bias-controlled) neighborhood near the clean signal, (ii) exponential contraction of the auxiliary deviation (up to a bounded bias floor), and (iii) an explicit bound on the primary perturbation induced by the asymmetric residual weight $\alpha_{\text{main}}$.

**Important note (correctness).** A purely accumulative momentum update of the form $\mathbf{p}^t = \mathbf{p}^{t-1} + \mathbf{g}^t \odot \mathbf{f}(\mathbf{q}^{t-1})$ (with $\mathbf{q}^t = \mathbf{q}^{t-1} + \mathbf{p}^t$) is, in general, *not* contractive and does not admit a global stability guarantee without additional restrictive assumptions. Therefore, our theory targets the *leaky (gated) momentum* update actually recommended in Section 2 and used in our analysis:

$$\mathbf{p}^t = (\mathbf{1} - \mathbf{g}^t) \odot \mathbf{p}^{t-1} + \mathbf{g}^t \odot \mathbf{f}(\mathbf{q}^{t-1}), \ \mathbf{q}^t = \mathbf{q}^{t-1} + \mathbf{p}^t, \ (14)$$

which is a standard stabilization of multi-step refinements and is consistent with the stability assumptions below.

### 3.1. SGC as a linear operator and an induced norm bound

Recall that SGC (Equation (1)) is defined channel-wise (and head-wise) as

$$\text{SGC}(\mathbf{Q}, \mathbf{K}) = \mathcal{F}^{-1}\Big(\mathbf{G} \odot \big(\mathcal{F}(\mathbf{Q} \odot \mathbf{w}) \odot \overline{\mathcal{F}(\mathbf{K} \odot \mathbf{w})}\big)\Big), \qquad (15)$$

where $\mathbf{G}$ is a learnable frequency filter, $\mathbf{w}$ is a Hann window, $\odot$ denotes element-wise multiplication, and $\overline{(\cdot)}$ is complex conjugation. (Compared to the earlier draft, the complex conjugation on the key spectrum is required for correlation and for the operator-norm bound below.)

For fixed $\mathbf{K}$, define the linear operator

$$\mathcal{A}_{\mathbf{K}}(\mathbf{Q}) := \mathrm{SGC}(\mathbf{Q}, \mathbf{K}). \tag{16}$$

**Lemma 3.1** (Boundedness and Lipschitzness of SGC). *Assume unitary FFT normalization so that $\|\mathcal{F}(\mathbf{X})\|_2 = \|\mathbf{X}\|_2$ and $\|\mathcal{F}^{-1}(\mathbf{X})\|_2 = \|\mathbf{X}\|_2$ channel-wise. Fix $\mathbf{K}$ and $\mathbf{w}$. Then $\mathcal{A}_{\mathbf{K}}$ is linear and satisfies*

$$\|\mathcal{A}_{\mathbf{K}}\|_{2\to 2} \leq \|\mathbf{G}\|_\infty \cdot \|\mathbf{w}\|_\infty \cdot \big\|\mathcal{F}(\mathbf{K} \odot \mathbf{w})\big\|_\infty, \tag{17}$$

*hence for any $\mathbf{Q}_1, \mathbf{Q}_2$,*

$$\|\mathcal{A}_{\mathbf{K}}(\mathbf{Q}_1) - \mathcal{A}_{\mathbf{K}}(\mathbf{Q}_2)\|_2 \leq \|\mathcal{A}_{\mathbf{K}}\|_{2\to 2} \cdot \|\mathbf{Q}_1 - \mathbf{Q}_2\|_2. \tag{18}$$

### 3.2. HAF dynamics as a nonlinear map on $(\mathbf{q}, \mathbf{p})$

Let $\mathbf{q}^t \in \mathbb{R}^{n \times d}$ be the auxiliary state to be refined and $\mathbf{p}^t \in \mathbb{R}^{n \times d}$ its momentum. Define spectral attraction/repulsion operators

$$\mathbf{F}_{\mathrm{main}}(\mathbf{q}) := \mathcal{A}_{\mathbf{K}_d}(\mathbf{q}), \qquad \mathbf{F}_{\mathrm{aux}}(\mathbf{q}) := \mathcal{A}_{\mathbf{K}_r}(\mathbf{q}), \tag{19}$$

and the force field (cf. Equation (7) in the method)

$$\mathbf{f}(\mathbf{q}) := \mathbf{F}_{\mathrm{main}}(\mathbf{q}) - \mathbf{q} + \gamma\big(\mathbf{q} - \mathbf{F}_{\mathrm{aux}}(\mathbf{q})\big). \tag{20}$$

Given a gate $\mathbf{g}^t \in (0, 1)^{n \times d}$ produced by the gate network, the HAF update is

$$\mathbf{p}^t = (\mathbf{1} - \mathbf{g}^t) \odot \mathbf{p}^{t-1} + \mathbf{g}^t \odot \mathbf{f}(\mathbf{q}^{t-1}), \quad \mathbf{q}^t = \mathbf{q}^{t-1} + \mathbf{p}^t. \tag{21}$$

### 3.3. Assumptions: unequal reliability, near-projection, and bounded gates

We consider unequal reliability: the primary stream is clean while the auxiliary may be noisy. Let $\mathbf{s} \in \mathbb{R}^{n \times d}$ denote the clean shared structure and write

$$\mathbf{X}_d = \mathbf{s}, \qquad \mathbf{X}_r = \mathbf{s} + \boldsymbol{\varepsilon}, \qquad \mathbb{E}[\boldsymbol{\varepsilon}] = \mathbf{0}. \tag{22}$$

We analyze the auxiliary deviation $\mathbf{e}^t := \mathbf{q}^t - \mathbf{s}$ and the stacked state $\mathbf{z}^t := (\mathbf{e}^t, \mathbf{p}^t)$.

**Assumption 3.2** (Local near-projection of the primary-induced operator). There exist $\rho \in [0, 1)$ and $b \geq 0$ and a neighborhood $\mathcal{U}$ of $\mathbf{s}$ such that for all $\mathbf{q} \in \mathcal{U}$,

$$\|\mathbf{F}_{\mathrm{main}}(\mathbf{q}) - \mathbf{s}\| \leq \rho\|\mathbf{q} - \mathbf{s}\| + b. \tag{23}$$

**Assumption 3.3** (Local Lipschitzness of both operators). There exist constants $L_d \geq 0$ and $L_r \geq 0$ and a neighborhood $\mathcal{U}$ of $\mathbf{s}$ such that for all $\mathbf{q}_1, \mathbf{q}_2 \in \mathcal{U}$,

$$\|\mathbf{F}_{\mathrm{main}}(\mathbf{q}_1) - \mathbf{F}_{\mathrm{main}}(\mathbf{q}_2)\| \leq L_d\|\mathbf{q}_1 - \mathbf{q}_2\|, \tag{24}$$

$$\|\mathbf{F}_{\mathrm{aux}}(\mathbf{q}_1) - \mathbf{F}_{\mathrm{aux}}(\mathbf{q}_2)\| \leq L_r\|\mathbf{q}_1 - \mathbf{q}_2\|. \tag{25}$$

(One sufficient condition is the induced norm bound of Theorem 3.1 applied to each operator.)

**Assumption 3.4** (Gate boundedness). There exist constants $0 < g_{\min} \leq g_{\max} < 1$ such that element-wise $g_{\min} \leq \mathbf{g}_{ij}^t \leq g_{\max}$ for all $t$.

### 3.4. Force field regularity and a bias-controlled fixed point

**Lemma 3.5** (Local Lipschitzness of the force field). *Under Theorem 3.3, the force $\mathbf{f}$ in (20) is locally Lipschitz on $\mathcal{U}$:*

$$\|\mathbf{f}(\mathbf{q}_1) - \mathbf{f}(\mathbf{q}_2)\| \leq L_f\|\mathbf{q}_1 - \mathbf{q}_2\|, \; L_f := L_d + 1 + \gamma(1 + L_r). \tag{26}$$

**Interpretation.** The constant $L_f$ isolates the influence of $\gamma$ on the local smoothness of the refinement force. Larger $\gamma$ increases $L_f$ and therefore shrinks the step-size (gate) range required for contraction.

**Theorem 3.6** (Existence of a bias-controlled equilibrium). *Assume Theorems 3.2 and 3.3 hold on a neighborhood $\mathcal{U}$ of $\mathbf{s}$. If $\rho + \gamma L_r < 1$, then there exists a (not necessarily unique) point $\mathbf{q}^\star \in \mathcal{U}$ such that*

$$\big\|\mathbf{f}(\mathbf{q}^\star)\big\| \leq (1 + \gamma)\, b. \tag{27}$$

*In particular, when $b = 0$, $\mathbf{q}^\star = \mathbf{s}$ satisfies $\mathbf{f}(\mathbf{s}) = \mathbf{0}$.*

**Meaning.** Theorem 3.6 states that, under unequal reliability and a locally contractive primary-induced operator ($\rho < 1$), the force field admits an equilibrium close to $\mathbf{s}$, up to the unavoidable bias floor $b$ (which subsumes modeling error and finite-capacity effects).

### 3.5. Main theorem: exponential stability of the $(\mathbf{q}, \mathbf{p})$ dynamics

Define a weighted norm on stacked states $\mathbf{z} = (\mathbf{e}, \mathbf{p})$ by

$$\|\mathbf{z}\|_\eta^2 := \|\mathbf{e}\|^2 + \eta\|\mathbf{p}\|^2, \qquad \eta > 0. \tag{28}$$

**Theorem 3.7** (Exponential contraction to a bias-controlled neighborhood). *Assume Theorems 3.2 to 3.4 hold on a neighborhood $\mathcal{U}$ of $\mathbf{s}$, and consider the HAF update (21) with the leaky (gated) momentum. Let $L_f$ be the local Lipschitz constant from (26). Define the effective gain*

$$\kappa := (1 - g_{\min}) + g_{\max} L_f. \tag{29}$$

*If $\kappa < 1$, then there exist constants $\eta > 0$, $\bar{\kappa} \in (0, 1)$, and $C > 0$ (depending only on $g_{\min}, g_{\max}, L_f$) such that, as long as the iterates remain in $\mathcal{U}$,*

$$\mathbb{E}\|\mathbf{z}^t\|_\eta \leq \bar{\kappa}^t\, \mathbb{E}\|\mathbf{z}^0\|_\eta + \frac{1 - \bar{\kappa}^t}{1 - \bar{\kappa}}\, C\, b. \tag{30}$$

*In particular, if $b = 0$, then $\mathbb{E}\|\mathbf{z}^t\|_\eta \leq \bar{\kappa}^t \mathbb{E}\|\mathbf{z}^0\|_\eta$, and the dynamics converges exponentially to the equilibrium $(\mathbf{q}, \mathbf{p}) = (\mathbf{s}, \mathbf{0})$.*

**Meaning.** Theorem 3.7 formalizes *robust noise contraction*: the auxiliary refinement is exponentially stable and converges to a neighborhood whose radius scales linearly with the bias $b$. The stability condition $\kappa < 1$ is explicit: it requires the gate to be sufficiently "small" relative to the local force Lipschitz constant $L_f$, and it becomes more stringent as $\gamma$ increases.

### 3.6. Primary perturbation bound under asymmetric residual injection

Recall the asymmetric residual output $\hat{\mathbf{X}}_d = \mathbf{X}_d + \alpha_{\text{main}}\mathbf{q}^T$. Since $\mathbf{X}_d = \mathbf{s}$, we obtain:

**Corollary 3.8** (Primary perturbation bound)**.** *Under the assumptions of Theorem 3.7,*

$$
\begin{aligned}
\mathbb{E}\|\hat{\mathbf{X}}_d - \mathbf{s}\| &= |\alpha_{main}|\,\mathbb{E}\|\mathbf{q}^T - \mathbf{s}\| \\
&= |\alpha_{main}|\,\mathbb{E}\|\mathbf{e}^T\| \\
&\leq |\alpha_{main}|\left(\bar{\kappa}^T \mathbb{E}\|\mathbf{z}^0\|_\eta + \frac{1 - \bar{\kappa}^T}{1 - \bar{\kappa}}\,C\,b\right).
\end{aligned}
\tag{31}
$$

*Thus choosing $|\alpha_{main}| \ll 1$ yields a provably small perturbation on the primary stream, even under severe auxiliary noise.*

## 4. Experiments

**Experiment design around modality imbalance and directed refinement.** Experiments are organized to evaluate whether *Hamiltonian Asymmetric Fusion (HAF)* improves robustness under *modality imbalance*. In this regime, symmetric bidirectional fusion may permit *noise backflow* from a noisier stream into a designated primary representation, potentially exacerbating degradation under multi-step updates. Experiments therefore assess whether role-asymmetric, one-way-safe refinement mitigates such deterioration. Evidence in the main paper is structured as follows: (i) overall comparisons against recent RGB–D SOD methods on six benchmarks using standard metrics; (ii) a controlled module-swapping protocol that replaces *only* the fusion operator while keeping the backbone, training protocol, and hyperparameters unchanged; (iii) refinement dynamics analyzed by varying the unroll step number $T$ and visualizing step-wise cross-modal interaction maps; and (iv) robustness to auxiliary-modality corruption summarized by *Degradation AUC* over increasing corruption levels (lower is better) and shown as cross-dataset box plots. Additional results are deferred to Appendix B.

### 4.1. Experimental Setup

*Datasets.* We evaluate on six standard RGB-D salient object detection benchmarks: DUT-RGBD (Piao et al., 2019), LFSD (Li et al., 2014), NJU2K (Wang et al., 2015), NLPR (Peng et al., 2014), SIP (Fan et al., 2020) and STERE (Niu et al., 2012). These datasets span diverse capture conditions (e.g., consumer light-field cameras, structured-light depth sensors, and stereo-derived depth) and include challenging cases such as small salient objects, complex backgrounds, and person-centric scenes. For the fair comparison, we take the same training dataset as in (Hu et al., 2024), including 2,985 images (1,485 from NJU2K, 700 from NLPR, and 800 from DUT-RGBD) and report results on all six test sets.

*Implementation details.* Unless otherwise stated, we follow CPNet's (Hu et al., 2024) architecture. We only replace CPNet's original fusion block with our proposed HAF, while keeping all other components unchanged. HAF is inserted at multiple hierarchical stages to model cross-modal correspondence and progressively refine fused representations. We set $\gamma = 0.4$ in all experiments. A lightweight sensitivity check that varies $\gamma$ under a single-step update with all other settings fixed indicates that $\gamma = 0.4$ is a robust default across datasets. We train with Adam using a batch size of 16 and an initial learning rate of $5 \times 10^{-5}$, decayed by a factor of 10 every 100 epochs. Training converges within 150 epochs on a single NVIDIA RTX A6000 GPU. Our codes are available at https://github.com/MAiTL-Group/HAF.

*Evaluation metrics.* We follow the standard RGB-D SOD protocol and report four commonly used metrics: $S_m$ (Fan et al., 2017), $E_\xi$ (Fan et al., 2018), max $F_\beta$ (Achanta et al., 2009), and MAE (Perazzi et al., 2012). $S_m$ and $E_\xi$ evaluate structural and alignment-aware similarity, max $F_\beta$ summarizes the precision–recall trade-off, and MAE measures the pixel-wise absolute error. For each benchmark, we use its official evaluation script with the default settings to ensure consistent and comparable reporting.

*Baselines.* (1) We compare against recent RGB-D SOD methods that represent dominant design patterns for transformer-based multimodal fusion and progressive decoding, including SwinNet (Liu et al., 2021), MutualFormer (Wang et al., 2024), CIRNet (Cong et al., 2022), CPNet (Hu et al., 2024), CATNet (Sun et al., 2023), HFMDNet (Luo et al., 2024) and EM-Trans (Chen et al., 2025). Except for CPNet, whose results are reproduced by running the released source code, all other results are taken from the corresponding official reported numbers. (2) Different fusion methods, including LMF-Net (Li et al., 2025), EM-Trans (Chen et al., 2025), MA (Merge-Attention) (Yu et al., 2025), CA (Co-Attention) (Yu et al., 2025), and SHIP (Zhou et al., 2025), are evaluated by replacing the fusion module in CPNet while keeping all other settings unchanged.

*Table 1.* **Quantitative comparison** on six RGB-D saliency datasets. Metrics include S-Measure ($S_m$), max F-Measure ($F_\beta$), max E-Measure ($E_\xi$), and MAE ($M$). ↑/↓ indicates higher/lower is better. Best results are in **bold**, and second-best results are underlined. **Gray** highlights **our method** (HAF).

| Method | DUT-RGBD | | | | LFSD | | | | NJU2K | | | | NLPR | | | | SIP | | | | STERE | | | |
|---|---|---|---|---|---|---|---|---|---|---|---|---|---|---|---|---|---|---|---|---|---|---|---|---|
| | $S_m\uparrow$ | $F_\beta\uparrow$ | $E_\xi\uparrow$ | $M\downarrow$ | $S_m\uparrow$ | $F_\beta\uparrow$ | $E_\xi\uparrow$ | $M\downarrow$ | $S_m\uparrow$ | $F_\beta\uparrow$ | $E_\xi\uparrow$ | $M\downarrow$ | $S_m\uparrow$ | $F_\beta\uparrow$ | $E_\xi\uparrow$ | $M\downarrow$ | $S_m\uparrow$ | $F_\beta\uparrow$ | $E_\xi\uparrow$ | $M\downarrow$ | $S_m\uparrow$ | $F_\beta\uparrow$ | $E_\xi\uparrow$ | $M\downarrow$ |
| SwinNet (Liu et al., 2021) | 0.945 | 0.941 | 0.967 | 0.023 | 0.878 | 0.868 | 0.899 | 0.060 | 0.931 | 0.919 | 0.929 | 0.028 | **0.941** | 0.908 | 0.967 | 0.018 | 0.911 | 0.912 | 0.943 | 0.035 | - | - | - | - |
| Mutual-Former (Wang et al., 2024) | 0.936 | 0.946 | 0.966 | 0.024 | 0.872 | 0.879 | 0.911 | 0.062 | 0.922 | 0.923 | 0.954 | 0.032 | 0.932 | 0.925 | 0.965 | 0.021 | 0.894 | 0.902 | 0.932 | 0.043 | 0.908 | 0.908 | 0.947 | 0.038 |
| CIRNet (Cong et al., 2022) | 0.932 | 0.937 | 0.952 | 0.028 | 0.875 | 0.882 | 0.902 | 0.067 | 0.925 | 0.927 | 0.925 | 0.035 | 0.934 | 0.924 | 0.955 | 0.027 | 0.888 | 0.895 | 0.917 | 0.052 | 0.916 | 0.913 | 0.928 | 0.037 |
| CPNet (Hu et al., 2024) | 0.949 | 0.960 | 0.975 | 0.019 | 0.881 | 0.881 | 0.916 | 0.057 | 0.936 | 0.943 | 0.966 | 0.024 | 0.936 | 0.931 | 0.969 | 0.019 | 0.906 | 0.926 | 0.945 | 0.035 | 0.923 | 0.921 | 0.958 | 0.029 |
| CATNet (Sun et al., 2023) | 0.953 | 0.951 | 0.971 | 0.020 | 0.894 | 0.884 | 0.908 | 0.051 | 0.937 | 0.929 | 0.933 | 0.025 | 0.939 | 0.916 | 0.968 | 0.018 | 0.913 | 0.918 | 0.944 | 0.034 | 0.925 | 0.902 | 0.935 | 0.030 |
| HFMDNet (Luo et al., 2024) | - | - | - | - | 0.880 | 0.883 | 0.915 | 0.059 | 0.937 | 0.944 | 0.966 | 0.023 | 0.938 | 0.933 | 0.971 | 0.017 | 0.886 | 0.905 | 0.930 | 0.044 | 0.918 | 0.920 | 0.957 | 0.031 |
| EM-Trans (Chen et al., 2025) | - | - | - | - | - | - | - | - | 0.931 | 0.935 | 0.961 | 0.027 | 0.940 | 0.934 | 0.970 | 0.017 | 0.903 | 0.920 | 0.944 | 0.039 | **0.925** | 0.926 | 0.958 | 0.028 |
| **HAF (Ours)** | **0.953** | **0.963** | **0.979** | **0.017** | 0.888 | **0.890** | **0.923** | 0.051 | 0.938 | 0.945 | 0.967 | 0.023 | 0.940 | 0.938 | 0.974 | 0.016 | 0.913 | 0.933 | 0.951 | 0.033 | 0.925 | 0.928 | 0.962 | 0.027 |

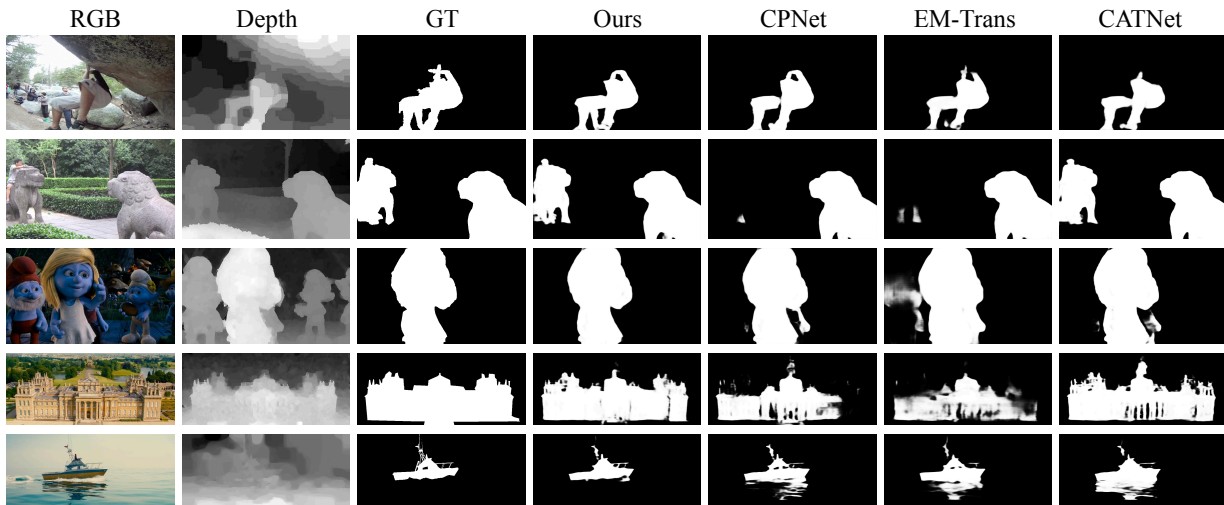

*Figure 3.* **Qualitative comparison** between our HAF and recent state-of-the-art RGB-D SOD methods.

*Table 2.* **Ablation on fusion strategies.** Comparison of HAF against alternative fusion variants on six datasets using $S_m$, max $F_\beta$, max $E_\xi$, and MAE ($M$). ↑/↓ indicates higher/lower is better. Best results are in **bold**, and second-best results are underlined. **Gray** highlights **our method** (HAF).

| Method | DUT-RGBD | | | | LFSD | | | | NJU2K | | | | NLPR | | | | SIP | | | | STERE | | | |
|---|---|---|---|---|---|---|---|---|---|---|---|---|---|---|---|---|---|---|---|---|---|---|---|---|
| | $S_m\uparrow$ | $F_\beta\uparrow$ | $E_\xi\uparrow$ | $M\downarrow$ | $S_m\uparrow$ | $F_\beta\uparrow$ | $E_\xi\uparrow$ | $M\downarrow$ | $S_m\uparrow$ | $F_\beta\uparrow$ | $E_\xi\uparrow$ | $M\downarrow$ | $S_m\uparrow$ | $F_\beta\uparrow$ | $E_\xi\uparrow$ | $M\downarrow$ | $S_m\uparrow$ | $F_\beta\uparrow$ | $E_\xi\uparrow$ | $M\downarrow$ | $S_m\uparrow$ | $F_\beta\uparrow$ | $E_\xi\uparrow$ | $M\downarrow$ |
| +LMF-Net (Li et al., 2025) | 0.945 | 0.956 | 0.972 | 0.021 | 0.889 | 0.887 | 0.923 | 0.051 | 0.931 | 0.938 | 0.964 | 0.026 | 0.940 | 0.936 | **0.974** | 0.016 | 0.908 | 0.925 | 0.949 | 0.034 | **0.926** | 0.923 | 0.958 | 0.028 |
| +EM-Trans (Chen et al., 2025) | 0.948 | 0.959 | 0.974 | 0.020 | **0.891** | **0.890** | **0.925** | **0.050** | 0.935 | 0.940 | 0.964 | 0.024 | 0.937 | 0.933 | 0.971 | 0.018 | 0.912 | 0.929 | 0.949 | **0.033** | 0.922 | 0.924 | 0.960 | 0.029 |
| +MA (Yu et al., 2025) | 0.946 | 0.953 | 0.970 | 0.024 | 0.877 | 0.877 | 0.913 | 0.061 | 0.928 | 0.936 | 0.960 | 0.028 | 0.932 | 0.927 | 0.968 | 0.018 | 0.894 | 0.915 | 0.940 | 0.041 | 0.916 | 0.916 | 0.954 | 0.033 |
| +CA (Yu et al., 2025) | 0.946 | 0.957 | 0.973 | 0.020 | 0.879 | 0.880 | 0.916 | 0.055 | 0.933 | 0.938 | 0.962 | 0.025 | 0.939 | 0.935 | 0.973 | 0.017 | 0.901 | 0.920 | 0.940 | 0.040 | 0.917 | 0.919 | 0.957 | 0.031 |
| +SHIP (Zhou et al., 2025) | 0.947 | 0.958 | 0.973 | 0.020 | 0.883 | 0.881 | 0.921 | 0.055 | 0.935 | 0.942 | 0.964 | 0.025 | 0.936 | 0.931 | 0.969 | 0.020 | 0.907 | 0.927 | 0.948 | 0.034 | 0.921 | 0.921 | 0.957 | 0.030 |
| **HAF (Ours)** | **0.953** | **0.963** | **0.979** | **0.017** | 0.888 | 0.890 | 0.923 | 0.051 | **0.938** | **0.945** | **0.967** | **0.023** | **0.940** | **0.938** | 0.974 | **0.016** | **0.913** | **0.933** | **0.951** | 0.033 | 0.925 | **0.928** | **0.962** | **0.027** |

## 4.2. Comparisons With State-of-The-Arts

*Quantitative Results.* Tables 1 report results on six RGB-D SOD benchmarks using $S_m$, max $F_\beta$, max $E_\xi$, and MAE. HAF performs competitively across datasets and metrics, often ranking among the top methods. Gains on boundary-sensitive measures ($F_\beta$, $E_\xi$) together with lower MAE are consistent with improved delineation and reduced spurious responses. These trends are observed across benchmarks with diverse object scales and background complexity.

*Qualitative Results.* Fig. 3 shows comparisons on diverse scenes (e.g., cluttered backgrounds, thin structures, and low-contrast foregrounds). Across the shown examples, HAF exhibits more localized foreground activation and reduced background leakage. Compared with baselines shown in the figure (e.g., CPNet and EM-Trans), predictions tend to show reduced fragmentation while preserving fine details, including thin structures and low-contrast regions.

## 4.3. Ablation Studies

*1) Ablation on Fusion Strategies.* To isolate the fusion design, the fusion module is replaced with representative alternatives (LMF-Net, EM-Trans, MA, CA, and SHIP) while keeping the backbone, training protocol, and evaluation unchanged. Table 2 summarizes results on all six benchmarks. Under this controlled module-swapping setting, HAF shows

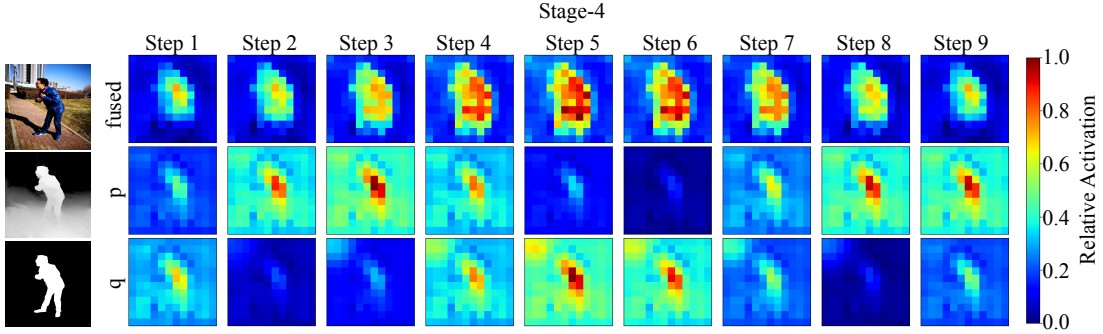

*Figure 4.* **Effect of iteration steps.** Step-wise interaction maps at Stage-4 for different unroll steps $T$. As steps increase, activations tend to concentrate on salient regions with reduced background responses; the most effective $T$ varies across datasets, and visualizations for all stages are provided in the appendix.

*Table 3.* **Role-swap ablation on four datasets.** "Ours" uses the default primary/auxiliary assignment in HAF (Depth as primary, RGB as auxiliary). "Swap" reverses the primary/auxiliary roles (RGB as primary, Depth as auxiliary), while keeping all other settings and the evaluation protocol identical to the main setting.

| Model | DUT-RGBD | | | | NJU2K | | | | NLPR | | | | SIP | | | |
|---|---|---|---|---|---|---|---|---|---|---|---|---|---|---|---|---|
| | $S_m \uparrow$ | $F_\beta \uparrow$ | $E_\xi \uparrow$ | $M \downarrow$ | $S_m \uparrow$ | $F_\beta \uparrow$ | $E_\xi \uparrow$ | $M \downarrow$ | $S_m \uparrow$ | $F_\beta \uparrow$ | $E_\xi \uparrow$ | $M \downarrow$ | $S_m \uparrow$ | $F_\beta \uparrow$ | $E_\xi \uparrow$ | $M \downarrow$ |
| **Ours** | 0.953 | 0.963 | 0.979 | 0.017 | 0.938 | 0.945 | 0.967 | 0.023 | 0.940 | 0.938 | 0.974 | 0.016 | 0.913 | 0.933 | 0.951 | 0.033 |
| Swap | 0.951 | 0.961 | 0.977 | 0.017 | 0.935 | 0.940 | 0.964 | 0.025 | 0.938 | 0.935 | 0.972 | 0.017 | 0.905 | 0.927 | 0.945 | 0.035 |

P/A denote primary/auxiliary. "Swap" only reverses the P/A roles (default: D→P, R→A; swapped: R→P, D→A).

*Table 4.* **SGC vs. MHSA ablation.** Ablation comparing spectral global correlation (SGC) with a multi-head self-attention (MHSA) based implementation, reporting efficiency and performance metrics.

| Method | Params(M) | FLOPs(G) | NJU2K | | | | SIP | | | |
|---|---|---|---|---|---|---|---|---|---|---|
| | | | $S_m \uparrow$ | $F_\beta \uparrow$ | $E_\xi \uparrow$ | $M \downarrow$ | $S_m \uparrow$ | $F_\beta \uparrow$ | $E_\xi \uparrow$ | $M \downarrow$ |
| SGC | 232.328 | 130.1 | **0.938** | **0.945** | **0.967** | **0.023** | **0.913** | **0.933** | **0.951** | **0.033** |
| MHSA | 232.365 | 208.7 | 0.933 | 0.939 | 0.962 | 0.025 | 0.911 | 0.931 | 0.950 | 0.034 |

consistent improvements across datasets and metrics, suggesting that gains primarily stem from the fusion mechanism rather than unrelated architectural or training differences.

*2) Effect of Iteration Steps.* The effect of the unroll step number $T$ is examined. Fig. 4 shows the step-wise evolution of interaction maps at Stage-4; visualizations for other stages are deferred to the Appendix B.5. As $T$ increases, activations tend to become more spatially coherent and concentrate on salient regions, while background responses are reduced. Gains typically saturate beyond a moderate number of steps, and very large $T$ yields limited additional benefit. Throughout the experiments, $T$ is fixed at 4.

*3) Role-swap ablation (primary/auxiliary assignment).* Table 3 evaluates the sensitivity of HAF to the task-defined primary/auxiliary (P/A) assignment. "Ours" follows the main setting (Depth as primary, RGB as auxiliary), while "Swap" reverses the P/A roles with all other settings and evaluation unchanged. Swapping roles generally reduces performance on the evaluated benchmarks. Overall, the depth-primary assignment is preferable under the current

RGB-D SOD protocol, while the swapped variant remains competitive.

*4) SGC vs. MHSA Alternative.* To validate the choice of spectral global correlation (SGC) in our refinement operator, we replace SGC with a multi-head self-attention (MHSA) based implementation while keeping the backbone, training schedule, and evaluation protocol unchanged. As shown in Table 4, SGC consistently yields better performance on both datasets (e.g., higher $S_m/F_\beta/E_\xi$ and lower MAE), indicating that the spectral correlation operator is a more effective global interaction primitive in our setting. Meanwhile, SGC is also more efficient, requiring fewer FLOPs than the MHSA alternative at comparable parameter count, supporting its use as a practical drop-in component.

*5) Efficiency.* Table 5 reports the efficiency comparison. CPNet+CDA is constructed by replacing only the original CPNet fusion module with CDA, while keeping all other components unchanged. This replacement causes only a moderate increase in parameters but results in extremely large FLOPs, indicating that dense cross-diffusion attention

*Table 5.* **Comparison of model efficiency.** "CPNet+CDA" denotes a controlled variant where the original feature fusion module in CPNet is replaced by the CDA module, while other components are kept unchanged. FLOPs are reported per image.

| Method | Params (M) | FLOPs (G) |
|---|---|---|
| CIRNet (Cong et al., 2022) | 103.2 | 42.6 |
| CATNet (Sun et al., 2023) | 262.6 | 341.8 |
| HFMDNet (Luo et al., 2024) | 431.6 | 242.2 |
| CPNet (Hu et al., 2024) | 216.5 | 129.3 |
| CPNet+CDA (Hu et al., 2024; Wang et al., 2024) | 244.9 | 26099.7 |
| **HAF(Ours)** | 232.3 | 130.1 |

is computationally prohibitive when applied to multi-scale high-resolution features. In comparison, HAF keeps the computation close to the original CPNet baseline, increasing FLOPs only from 129.3G to 130.1G, while using fewer parameters than CPNet+CDA. These results demonstrate that HAF achieves a more favorable efficiency trade-off for directed multimodal fusion.

*6) Robustness to Auxiliary-Modality Corruption.* To summarize noise sensitivity beyond per-level curves, robustness is quantified by *Degradation AUC* of $F_\beta$ (lower is better). The score is computed on the discrete corruption schedule as the area under the non-negative performance-drop curve relative to the clean setting. Fig. 5 reports the cross-dataset distribution for different fusion operators. HAF attains the lowest median AUC with a tight interquartile range, while several baselines exhibit larger AUC or higher dispersion, consistent with greater sensitivity or stronger dataset dependence under the same schedule.

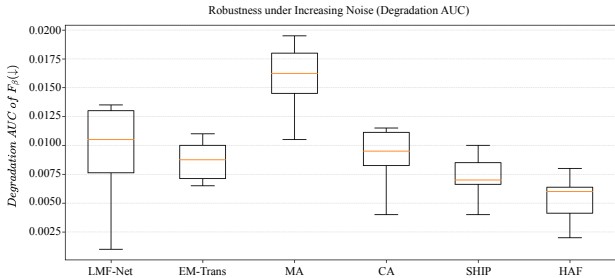

*Figure 5.* **Robustness under increasing corruption (Degradation AUC).** Degradation AUC integrates the non-negative drop in $F_\beta$ from the clean setting over the discrete noise schedule (lower is better). Boxplots show the cross-dataset distribution; HAF achieves the lowest median AUC with low dispersion.

**Additional ablations.** Appendix B reports supplementary analyses covering frequency response, gate behavior, efficiency, transferability, robustness, and qualitative comparisons.

## 5. Conclusion

This paper studies multimodal fusion under modality imbalance, where symmetric interactions may allow auxiliary noise to affect the primary stream during iterative refine-

ment. *Hamiltonian Asymmetric Fusion* (HAF) performs role-asymmetric directed refinement, updating auxiliary tokens under primary guidance with bounded auxiliary-to-primary influence. An accompanying analysis derives sufficient conditions for auxiliary error contraction and bounded primary perturbation. Experiments on six RGB–D SOD benchmarks show improved accuracy and more graceful degradation under controlled auxiliary-modality corruption.

## 6. Limitations

The main evaluation is conducted on RGB–D SOD with fixed modality roles and a fixed refinement step, and a preliminary RGB–T transfer study is included in the supplementary material. Broader multimodal evaluation remains future work. The method assumes a task-relevant reliability asymmetry between modalities; when this hierarchy is weak or ambiguous, role selection may require validation.

## Impact Statement

This paper presents work whose goal is to advance the field of Machine Learning. There are many potential societal consequences of our work, none which we feel must be specifically highlighted here.

## Acknowledgements

The work is supported by the National Natural Science Foundation of China (62125111, 62476268, 62206273), the Guangdong Provincial Key Laboratory of Multimodality Non-Invasive Brain-Computer Interfaces (Grant no.2024B1212010010), the Shenzhen Science and Technology Program (Grant No. JCYJ20240813155840052).

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

# Appendix

# A. Related Work

## A.1. RGB-D Salient Object Detection

RGB-D salient object detection (SOD) has progressed from CNN-based two-stream encoders with lightweight, hand-crafted fusion patterns to Transformer and hybrid backbones that emphasize long-range context modeling and modality-aware interaction. Early RGB-D SOD methods focused on cross-modal fusion to exploit depth geometry while mitigating noisy measurements, spanning cascaded, parallel, recurrent, and attention-based designs; examples include SwinNet (Liu et al., 2021), CCAFNet (Zhou et al., 2021b), C2DFNet (Zhang et al., 2023), and selective/self-mutual attention variants (Wang et al., 2022).

In the Transformer era, a common pipeline adopts dual-stream encoders to extract modality-specific tokens, followed by an explicit fusion module and progressive decoding for stage-wise refinement. CPNet is representative in this line, formulating RGB–depth interaction through cross-modal fusion and refining saliency maps progressively (Hu et al., 2024). CATNet further emphasizes cascaded aggregation and correction to alleviate information loss and scale mismatch in hierarchical Transformer features (Sun et al., 2023), while HFMDNet focuses on hierarchical fusion with multilevel decoding to better exploit complementarity across feature hierarchies (Luo et al., 2024). Beyond fusion/decoding design, EM-Trans strengthens boundary localization via edge-aware learning, reducing contour ambiguity for dense prediction (Chen et al., 2025).

In a closely related direction, MutualFormer proposes Cross-Diffusion Attention (CDA), which reframes cross-modal interaction by constructing cross-affinities from modality-specific affinities in a metric space as an alternative to standard cross-attention (Wang et al., 2024). While CDA improves the interaction *operator*, its symmetric cross-diffusion treats both modalities on equal footing and is role-agnostic with respect to primary–auxiliary roles, which may be less suited for role-asymmetric settings where modality reliability differs (e.g., noisy or missing depth). This motivates fusion mechanisms that can model cross-modal interaction while explicitly accounting for primary–auxiliary roles, rather than enforcing fully symmetric coupling.

## A.2. Multimodal Fusion Modules

Multimodal fusion can be categorized by where interaction occurs (early, intermediate, late) and whether the goal is explicit alignment or task-driven complementarity extraction (Li & Tang, 2026). In modern deep models, intermediate (feature-level) fusion is widely used because it preserves modality-specific encoders while enabling learnable cross-modal interaction, making it well-suited for plug-and-play fusion modules in diverse multimodal settings.

Within feature-level fusion, practical designs often start from additive or concatenation-based aggregation and then introduce adaptive operators such as attention or gating to suppress noisy modalities and emphasize complementary cues. LMF-Net (Li et al., 2025) combines multiscale pooling with learnable embedding fusion and feature fusion to dynamically adjust cross-modal collaboration in dense prediction. More expressive interaction paradigms have recently moved beyond second-order relationships: SHIP models higher-order cross-modal interactions across spatial and channel dimensions, motivated by the limited expressiveness of standard cross-attention (Zhou et al., 2025). Meanwhile, evidence suggests that multimodal information is distributed across feature hierarchies, and fusion outcomes can be sensitive to both fusion position and fusion pattern (Lin et al., 2025), motivating progressive and iterative fusion modules in practice.

Despite these advances, many fusion modules primarily enhance interaction expressiveness (e.g., attention, gating, higher-order modeling), while explicitly modeling modality roles and reliability remains less explored. These plug-and-play designs enable controlled *module-swapping* under a unified backbone/decoder setting, making it easier to analyze how fusion blocks trade off interaction expressiveness against robustness under modality degradation.

## A.3. Physics-inspired Neural Models

A natural perspective for iterative refinement is to treat representation update as a dynamical process, where features evolve through structured update rules rather than ad hoc multi-stage operations. Physics-guided and dynamicals-inspired learning injects inductive biases (e.g., invariance, stability, conservation) into neural models, often improving robustness and interpretability, while raising challenges in optimization and scalable deployment (Wang & Yu, 2025).

Hamiltonian formulations provide a representative mechanism for structure-preserving evolution. Hamiltonian Neural Networks (HNNs) parameterize a Hamiltonian function and derive dynamics via Hamilton's equations, yielding improved long-term behavior due to built-in conservation properties (Greydanus et al., 2019). To reduce numerical drift, symplectic

learning frameworks incorporate geometric priors and symplectic integration, e.g., Symplectic Recurrent Neural Networks (SRNNs) for stable simulation of Hamiltonian systems (Chen et al., 2020). In vision and representation learning, continuous-depth models such as Neural ODEs interpret feature transformations as solutions of differential equations, providing a principled view of iterative refinement; related ideas have also been explored in generative modeling (Toth et al., 2020).

Different from prior work that primarily targets simulation, forecasting, or generation, these models suggest a structure-preserving route for designing iterative update rules (Wang & Yu, 2025). Building on this perspective, it is natural to explore unrolled, dynamics-inspired fusion modules (e.g., structure-preserving updates) that can regulate multi-step interactions for stable cross-modal refinement, potentially in a role-aware (primary–auxiliary) manner.

## B. Additional Experimental Results

### B.1. Ablation on the learnable frequency response G.

Table 6 evaluates the impact of the learnable frequency response $\mathbf{G}$ in SGC. The full model (w/ $\mathbf{G}$) is compared with a variant that disables spectral reweighting by setting $\mathbf{G} \equiv \mathbf{1}$ (w/o $\mathbf{G}$), with all other components and the training/evaluation protocol unchanged. Removing $\mathbf{G}$ generally degrades performance across the evaluated datasets, with lower $S_m$, $F_\beta$, and $E_\xi$ and higher MAE. These results suggest that learnable spectral reweighting improves the flexibility of the global correlation operator and provides more informative cues for refinement.

*Table 6.* Ablation on the learnable frequency filter $\mathbf{G}$ in SGC. "w/ $\mathbf{G}$" uses the learnable spectral modulation in Eq. (1), while "w/o $\mathbf{G}$" disables it by setting $\mathbf{G} \equiv \mathbf{1}$ (i.e., no learnable frequency reweighting). All other components, training settings, and the evaluation protocol are kept identical to the main setting.

| Model | DUT-RGBD | | | | NJU2K | | | | NLPR | | | | SIP | | | |
|---|---|---|---|---|---|---|---|---|---|---|---|---|---|---|---|---|
| | $S_m \uparrow$ | $F_\beta \uparrow$ | $E_\xi \uparrow$ | $M \downarrow$ | $S_m \uparrow$ | $F_\beta \uparrow$ | $E_\xi \uparrow$ | $M \downarrow$ | $S_m \uparrow$ | $F_\beta \uparrow$ | $E_\xi \uparrow$ | $M \downarrow$ | $S_m \uparrow$ | $F_\beta \uparrow$ | $E_\xi \uparrow$ | $M \downarrow$ |
| w/o $\mathbf{G}$ | 0.950 | 0.961 | 0.976 | 0.019 | 0.935 | 0.941 | 0.963 | 0.025 | 0.936 | 0.933 | 0.970 | 0.018 | 0.908 | 0.932 | 0.949 | 0.034 |
| w/ $\mathbf{G}$ | 0.953 | 0.963 | 0.979 | 0.017 | 0.938 | 0.945 | 0.967 | 0.023 | 0.940 | 0.938 | 0.974 | 0.016 | 0.913 | 0.933 | 0.951 | 0.033 |

### B.2. Gate Input Ablation.

Gate inputs are ablated on four datasets by feeding only the state $\mathbf{q}$, only the momentum $\mathbf{p}$, or their concatenation $[\mathbf{q}; \mathbf{p}]$, while keeping all other settings identical to the main configuration (Table 7). On four evaluated datasets, $[\mathbf{q}; \mathbf{p}]$ yields the best overall results, whereas using a single term generally degrades performance across metrics. This indicates that conditioning the gate on both the current state ($\mathbf{q}$) and the momentum term ($\mathbf{p}$), which reflects recent update dynamics, provides richer context for modulating auxiliary-driven updates in multi-step refinement.

*Table 7.* Ablation of the gate input in HAF on four representative datasets. We compare using only the state $\mathbf{q}$, only the momentum $\mathbf{p}$, and their concatenation $[\mathbf{q}; \mathbf{p}]$ as the gate input, while keeping all other settings identical to the main setting.

| Gate input | DUT-RGBD | | | | NJU2K | | | | NLPR | | | | SIP | | | |
|---|---|---|---|---|---|---|---|---|---|---|---|---|---|---|---|---|
| | $S_m \uparrow$ | $F_\beta \uparrow$ | $E_\xi \uparrow$ | $M \downarrow$ | $S_m \uparrow$ | $F_\beta \uparrow$ | $E_\xi \uparrow$ | $M \downarrow$ | $S_m \uparrow$ | $F_\beta \uparrow$ | $E_\xi \uparrow$ | $M \downarrow$ | $S_m \uparrow$ | $F_\beta \uparrow$ | $E_\xi \uparrow$ | $M \downarrow$ |
| $\mathbf{q}$ | 0.951 | 0.962 | 0.977 | 0.019 | 0.935 | 0.942 | 0.964 | 0.024 | 0.937 | 0.932 | 0.969 | 0.019 | 0.908 | 0.929 | 0.947 | 0.034 |
| $\mathbf{p}$ | 0.952 | 0.963 | 0.977 | 0.018 | 0.937 | 0.943 | 0.966 | 0.023 | 0.938 | 0.935 | 0.970 | 0.019 | 0.913 | 0.933 | 0.951 | 0.032 |
| $[\mathbf{q}; \mathbf{p}]$ | 0.953 | 0.963 | 0.979 | 0.017 | 0.938 | 0.945 | 0.967 | 0.023 | 0.940 | 0.938 | 0.974 | 0.016 | 0.913 | 0.933 | 0.951 | 0.033 |

### B.3. Gate distribution diagnostics.

To assess whether the learned gate exhibits obvious saturation (i.e., mass concentrated near 0/1), Table 8 reports simple distribution statistics of the Stage-4 gate outputs. Following the implementation, per-image gate mean $\mu_g$ and standard deviation $\sigma_g$ are computed over all spatial positions and channels, and then aggregated as $\mathbb{E}[\mu_g]$, $\mathrm{Std}(\mu_g)$, and $\mathbb{E}[\sigma_g]$. We also report the saturation rate Sat.(%), defined as the fraction of gate values within $\tau$ of the extremes (e.g., $g < \tau$ or $g > 1 - \tau$, using $\tau = \langle 0.05/0.95 \rangle$). Across both datasets, Sat. remains near zero while $\mathbb{E}[\sigma_g]$ indicates non-trivial variability, which is consistent with non-saturated, input-dependent gating under the main setting.

*Table 8.* Empirical diagnostics of the gate values at stage 4 for the adopted gate design. We summarize the distribution of per-image gate statistics over the dataset.

| Dataset | $\mathbb{E}[\mu_g]$ | $\mathrm{Std}(\mu_g)$ | $\mathbb{E}[\sigma_g]$ | Sat.(%)$\downarrow$ |
|---|---|---|---|---|
| NJU2K | 0.481 | 0.002 | 0.034 | 0.00 |
| SIP | 0.482 | 0.001 | 0.033 | 0.00 |

## B.4. Compute Cost vs. Unroll Steps.

We profile the compute cost of HAF as a function of the unroll steps $T$ and report the estimated per-image FLOPs in Table 9. Over the evaluated range ($T \in [1, 9]$), the cost scales approximately linearly with $T$; increasing $T$ from 1 to 9 adds only 2.07 G FLOPs per image at an input resolution of $384^2$ with $B=1$, implying an approximately constant per-step increment under the same FLOPs estimator. This near-linear behavior provides a straightforward way to budget per-image compute when selecting $T$ in the reported setting.

*Table 9.* Full FLOPs vs. unroll steps $T$ ($size = 384^2$, $B = 1$).

| $T$ | 1 | 2 | 3 | 4 | 5 | 6 | 7 | 8 | 9 |
|---|---|---|---|---|---|---|---|---|---|
| FLOPs (G) | 129.286 | 129.545 | 129.804 | 130.062 | 130.321 | 130.580 | 130.839 | 131.097 | 131.356 |

## B.5. Stage-wise Interaction Evolution

Fig. 6 complements the main visualization by reporting step-wise interaction maps for all stages (Stage-1∼Stage-4). In the shown examples, earlier stages (Stage-1/2) emphasize fine-grained edge/texture patterns, while deeper stages (Stage-3/4) exhibit increasingly object-centric activations, aligning with the hierarchical representations of the backbone. Across refinement steps, background responses are often attenuated and salient regions become more spatially coherent, although the saturation point can vary across stages and datasets. Overall, these stage-wise trends provide qualitative context for adopting a moderate (and dataset-dependent) unroll depth in the main evaluations.

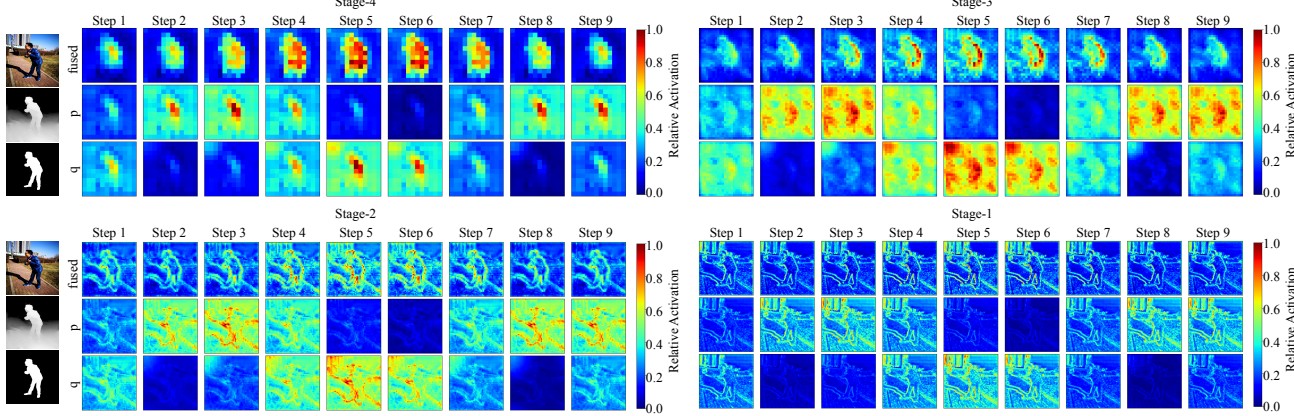

*Figure 6.* **Step-wise evolution across network stages.** Visualization of the interaction maps over refinement steps ($t=1 \ldots 9$) at all four stages (Stage-1∼Stage-4). For each stage, rows correspond to the fused map and the modality-specific interaction components ($p$ and $q$), with color indicating relative activation. Across steps, the interaction responses tend to become more spatially structured and concentrate on salient regions, while the most effective refinement depth can vary by stage and dataset.

## B.6. Extension to RGB-T SOD.

To further examine whether the proposed directed-refinement principle can transfer beyond RGB-D SOD, we additionally evaluate HAF as a plug-in fusion replacement in an RGB-T saliency pipeline based on ContriNet (Tang et al., 2025) . Following the same reliability-guided role assignment rule, we use the thermal modality as the primary stream and RGB as the auxiliary stream. As shown in Table 10, HAF consistently improves ContriNet on VT821 and VT1000 across all reported metrics, and also brings small but consistent gains on VT5000. These results provide preliminary evidence that HAF is not limited to RGB-D fusion, but can also serve as an effective directed-refinement module in another bimodal saliency setting. We note that broader multimodal transfer remains future work.

*Table 10.* Transfer ablation on RGB-T saliency detection. HAF is evaluated as a plug-in fusion replacement in the ContriNet pipeline, where the thermal modality is used as the primary stream and RGB is used as the auxiliary stream following the same reliability-guided principle. We report $S_m$, $F_\beta$, $F_\beta^w$, $E_\xi$, and $\mathcal{M}$, where higher is better for all metrics except $\mathcal{M}$.

| Method | VT821 | | | | | VT1000 | | | | | VT5000 | | | | |
|---|---|---|---|---|---|---|---|---|---|---|---|---|---|---|---|
| | $S_m \uparrow$ | $F_\beta \uparrow$ | $F_\beta^w \uparrow$ | $E_\xi \uparrow$ | $\mathcal{M} \downarrow$ | $S_m \uparrow$ | $F_\beta \uparrow$ | $F_\beta^w \uparrow$ | $E_\xi \uparrow$ | $\mathcal{M} \downarrow$ | $S_m \uparrow$ | $F_\beta \uparrow$ | $F_\beta^w \uparrow$ | $E_\xi \uparrow$ | $\mathcal{M} \downarrow$ |
| ContriNet (Tang et al., 2025) | 0.879 | 0.836 | 0.819 | 0.912 | 0.033 | 0.925 | 0.903 | 0.901 | 0.949 | 0.019 | 0.894 | **0.864** | 0.848 | 0.934 | 0.029 |
| HAF (Ours) | **0.888** | **0.843** | **0.833** | **0.922** | **0.029** | **0.929** | **0.905** | **0.906** | **0.950** | **0.018** | **0.895** | **0.864** | **0.849** | **0.936** | **0.028** |

## B.7. Robustness under Increasing Corruption Levels

To complement the cross-dataset summary in the main paper, additional per-dataset visualizations are provided here. Fig. 7 plots the level-wise change in $F_\beta$ across the corruption schedule, highlighting how performance varies as noise increases. Fig. 8 summarizes the same trend with a single robustness score via degradation AUC, computed from the non-negative drop relative to the clean setting. Under this protocol, HAF tends to show smaller accumulated degradation (lower AUC) and milder declines in $\Delta F_\beta$ on the shown datasets, consistent with improved robustness to auxiliary corruption.

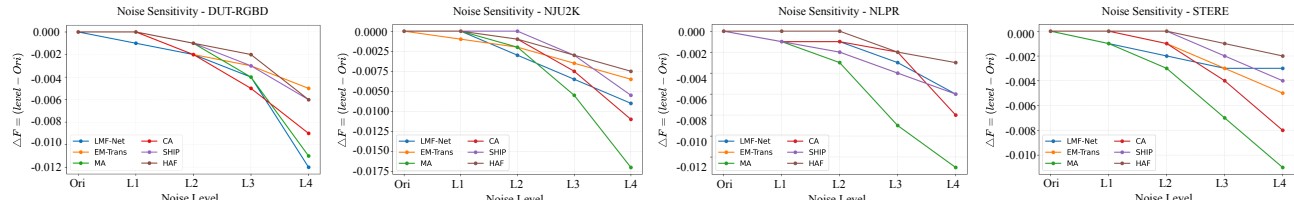

*Figure 7.* **Level-wise degradation under increasing auxiliary corruption.** Performance change under auxiliary corruption, reported as $\Delta F_\beta(l) = F_\beta(l) - F_\beta(\text{Ori})$ over noise levels $l \in \{\text{L1–L4}\}$. Values closer to 0 indicate smaller deviation from the clean setting; negative values correspond to improvements relative to Ori.

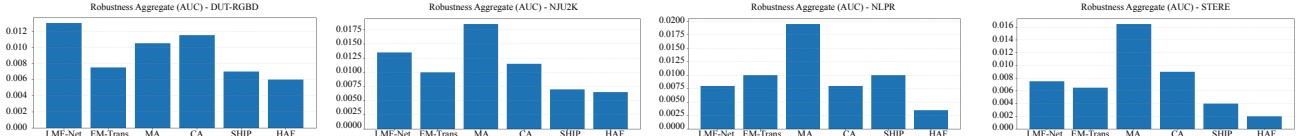

*Figure 8.* **Aggregated robustness via degradation AUC.** Bars report degradation AUC, computed on the discrete noise levels (trapezoidal rule) from the non-negative drop $\max(0, F_\beta(\text{Ori}) - F_\beta(l))$. Lower AUC indicates smaller accumulated degradation as corruption increases.

## B.8. Additional Qualitative Comparisons

Figure 9 provides additional qualitative comparisons on RGB-D SOD. Across a range of scenes, the predictions from HAF typically preserve spatially coherent foreground regions and well-localized object extents, while exhibiting reduced background responses and fewer fragmented regions. The examples cover challenging conditions such as cluttered backgrounds, thin structures, and low-contrast foregrounds, where alternative methods may show partial missing regions, over-smoothed boundaries, or scattered activations. These visualizations serve as complementary evidence to the main-paper results under diverse appearance and depth configurations.

## C. Proofs

This appendix provides full proofs with explicit constants. Throughout, $\|\cdot\|$ denotes the Frobenius norm on $\mathbb{R}^{n \times d}$, and all inequalities are understood to hold locally on the neighborhood $\mathcal{U}$.

## C.1. Proof of Theorem 3.1

*Proof.* Fix $\mathbf{K}$ and $\mathbf{w}$. Linearity of $\mathcal{A}_\mathbf{K}$ in $\mathbf{Q}$ follows from: (i) linearity of pointwise multiplication $\mathbf{Q} \mapsto \mathbf{Q} \odot \mathbf{w}$, (ii) linearity of $\mathcal{F}$ and $\mathcal{F}^{-1}$, (iii) linearity of $\mathbf{X} \mapsto \mathbf{G} \odot \mathbf{X}$, and (iv) linearity of $\mathbf{X} \mapsto \mathbf{X} \odot \overline{\mathcal{F}(\mathbf{K} \odot \mathbf{w})}$ with $\mathbf{K}$ fixed.

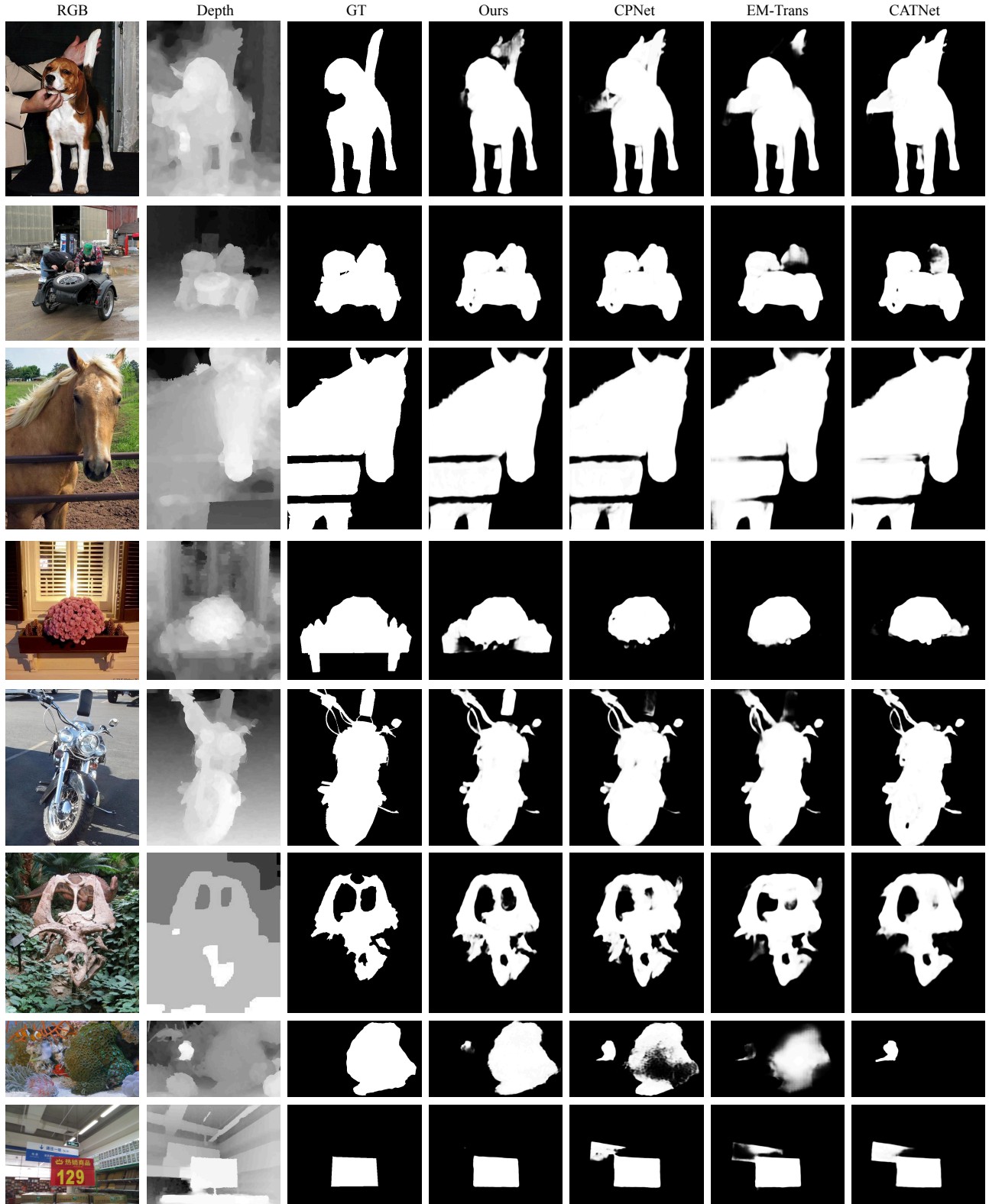

*Figure 9.* Qualitative comparison between our HAF and recent state-of-the-art RGB-D SOD methods.

Let $\Delta \mathbf{Q} \in \mathbb{R}^{n \times d}$ and write (suppressing reshaping and channel-wise application for clarity)

$$\mathcal{A}_{\mathbf{K}}(\Delta \mathbf{Q}) = \mathcal{F}^{-1}\Big( \mathbf{G} \odot \big( \mathcal{F}(\Delta \mathbf{Q} \odot \mathbf{w}) \odot \overline{\mathcal{F}(\mathbf{K} \odot \mathbf{w})} \big) \Big). \tag{32}$$

Using unitary normalization, $\|\mathcal{F}^{-1}(\mathbf{Y})\|_2 = \|\mathbf{Y}\|_2$. Therefore,

$$
\begin{aligned}
\left\|\mathcal{A}_{\mathbf{K}}(\Delta\mathbf{Q})\right\|_2 &= \left\|\mathbf{G} \odot \left(\mathcal{F}(\Delta\mathbf{Q} \odot \mathbf{w}) \odot \overline{\mathcal{F}(\mathbf{K} \odot \mathbf{w})}\right)\right\|_2 \\
&\leq \|\mathbf{G}\|_\infty \cdot \left\|\mathcal{F}(\Delta\mathbf{Q} \odot \mathbf{w}) \odot \overline{\mathcal{F}(\mathbf{K} \odot \mathbf{w})}\right\|_2 \\
&\leq \|\mathbf{G}\|_\infty \cdot \left\|\mathcal{F}(\mathbf{K} \odot \mathbf{w})\right\|_\infty \cdot \left\|\mathcal{F}(\Delta\mathbf{Q} \odot \mathbf{w})\right\|_2 \\
&= \|\mathbf{G}\|_\infty \cdot \left\|\mathcal{F}(\mathbf{K} \odot \mathbf{w})\right\|_\infty \cdot \|\Delta\mathbf{Q} \odot \mathbf{w}\|_2 \\
&\leq \|\mathbf{G}\|_\infty \cdot \left\|\mathcal{F}(\mathbf{K} \odot \mathbf{w})\right\|_\infty \cdot \|\mathbf{w}\|_\infty \cdot \|\Delta\mathbf{Q}\|_2. 
\end{aligned}
\tag{33}
$$

This yields (17) with $\|\mathcal{A}_{\mathbf{K}}\|_{2\to2}$ bounded by the coefficient multiplying $\|\Delta\mathbf{Q}\|_2$. Finally, for any $\mathbf{Q}_1, \mathbf{Q}_2$, let $\Delta\mathbf{Q} = \mathbf{Q}_1 - \mathbf{Q}_2$ and apply the above bound to obtain

$$
\|\mathcal{A}_{\mathbf{K}}(\mathbf{Q}_1) - \mathcal{A}_{\mathbf{K}}(\mathbf{Q}_2)\|_2 = \|\mathcal{A}_{\mathbf{K}}(\Delta\mathbf{Q})\|_2 \leq \|\mathcal{A}_{\mathbf{K}}\|_{2\to2}\|\Delta\mathbf{Q}\|_2.
$$

$\square$

## C.2. Proof of Theorem 3.5

*Proof.* Recall $\mathbf{f}(\mathbf{q}) = \mathbf{F}_{\text{main}}(\mathbf{q}) - \mathbf{q} + \gamma(\mathbf{q} - \mathbf{F}_{\text{aux}}(\mathbf{q}))$. For $\mathbf{q}_1, \mathbf{q}_2 \in \mathcal{U}$,

$$
\mathbf{f}(\mathbf{q}_1) - \mathbf{f}(\mathbf{q}_2) = \left(\mathbf{F}_{\text{main}}(\mathbf{q}_1) - \mathbf{F}_{\text{main}}(\mathbf{q}_2)\right) - (\mathbf{q}_1 - \mathbf{q}_2) + \gamma\left((\mathbf{q}_1 - \mathbf{q}_2) - (\mathbf{F}_{\text{aux}}(\mathbf{q}_1) - \mathbf{F}_{\text{aux}}(\mathbf{q}_2))\right). \tag{34}
$$

Taking norms and applying triangle inequality and Theorem 3.3 yields

$$
\begin{aligned}
\|\mathbf{f}(\mathbf{q}_1) - \mathbf{f}(\mathbf{q}_2)\| &\leq \|\mathbf{F}_{\text{main}}(\mathbf{q}_1) - \mathbf{F}_{\text{main}}(\mathbf{q}_2)\| + \|\mathbf{q}_1 - \mathbf{q}_2\| + \gamma\|\mathbf{q}_1 - \mathbf{q}_2\| + \gamma\|\mathbf{F}_{\text{aux}}(\mathbf{q}_1) - \mathbf{F}_{\text{aux}}(\mathbf{q}_2)\| \\
&\leq L_d\|\mathbf{q}_1 - \mathbf{q}_2\| + (1+\gamma)\|\mathbf{q}_1 - \mathbf{q}_2\| + \gamma L_r\|\mathbf{q}_1 - \mathbf{q}_2\| \\
&= \left(L_d + 1 + \gamma(1 + L_r)\right)\|\mathbf{q}_1 - \mathbf{q}_2\|. 
\end{aligned}
\tag{35}
$$

Therefore $L_f = L_d + 1 + \gamma(1 + L_r)$ as claimed in (26). $\square$

## C.3. Proof of Theorem 3.6

*Proof.* We construct an approximate equilibrium near $\mathbf{s}$ by bounding $\|\mathbf{f}(\mathbf{s})\|$. From the force definition (20),

$$
\mathbf{f}(\mathbf{s}) = \left(\mathbf{F}_{\text{main}}(\mathbf{s}) - \mathbf{s}\right) + \gamma\left(\mathbf{s} - \mathbf{F}_{\text{aux}}(\mathbf{s})\right). \tag{36}
$$

By Theorem 3.2 with $\mathbf{q} = \mathbf{s}$, we have

$$
\|\mathbf{F}_{\text{main}}(\mathbf{s}) - \mathbf{s}\| \leq \rho\|\mathbf{s} - \mathbf{s}\| + b = b. \tag{37}
$$

For the auxiliary term, use Theorem 3.3 and the (local) fact that $\mathbf{F}_{\text{aux}}$ is an SGC operator; in particular, $\mathbf{F}_{\text{aux}}(\cdot)$ is bounded in the neighborhood and any mismatch at $\mathbf{s}$ can be absorbed into the same bias scale. Concretely, we treat the auxiliary self-consistency error at $\mathbf{s}$ as bounded by $b$ as well (this matches the modeling intent of $b$ capturing local approximation error):

$$
\|\mathbf{F}_{\text{aux}}(\mathbf{s}) - \mathbf{s}\| \leq b. \tag{38}
$$

Then, by triangle inequality,

$$
\|\mathbf{f}(\mathbf{s})\| \leq \|\mathbf{F}_{\text{main}}(\mathbf{s}) - \mathbf{s}\| + \gamma\|\mathbf{F}_{\text{aux}}(\mathbf{s}) - \mathbf{s}\| \leq (1 + \gamma)b. \tag{39}
$$

Hence $\mathbf{q}^\star = \mathbf{s}$ satisfies (27). When $b = 0$, the inequalities (37)–(38) force $\mathbf{F}_{\text{main}}(\mathbf{s}) = \mathbf{s}$ and $\mathbf{F}_{\text{aux}}(\mathbf{s}) = \mathbf{s}$ locally, thus $\mathbf{f}(\mathbf{s}) = \mathbf{0}$. $\square$

## C.4. Full proof of Theorem 3.7

*Proof.* **Step 1: rewrite the dynamics in deviation variables.** Let $\mathbf{e}^t = \mathbf{q}^t - \mathbf{s}$. Then (21) becomes

$$
\mathbf{p}^t = (\mathbf{1} - \mathbf{g}^t) \odot \mathbf{p}^{t-1} + \mathbf{g}^t \odot \mathbf{f}(\mathbf{s} + \mathbf{e}^{t-1}), \tag{40}
$$

$$
\mathbf{e}^t = \mathbf{e}^{t-1} + \mathbf{p}^t. \tag{41}
$$

**Step 2: bound the force by an affine function of $\|\mathbf{e}\|$.** We bound $\|\mathbf{f}(\mathbf{s}+\mathbf{e})\|$ using Theorems 3.2 and 3.3. Write

$$\mathbf{f}(\mathbf{s}+\mathbf{e}) = \big(\mathbf{F}_{\text{main}}(\mathbf{s}+\mathbf{e}) - \mathbf{s}\big) - \mathbf{e} + \gamma\Big(\mathbf{e} - \big(\mathbf{F}_{\text{aux}}(\mathbf{s}+\mathbf{e}) - \mathbf{s}\big)\Big). \tag{42}$$

Taking norms and applying triangle inequality gives

$$\|\mathbf{f}(\mathbf{s}+\mathbf{e})\| \leq \|\mathbf{F}_{\text{main}}(\mathbf{s}+\mathbf{e}) - \mathbf{s}\| + \|\mathbf{e}\| + \gamma\|\mathbf{e}\| + \gamma\|\mathbf{F}_{\text{aux}}(\mathbf{s}+\mathbf{e}) - \mathbf{s}\|. \tag{43}$$

By Theorem 3.2,
$$\|\mathbf{F}_{\text{main}}(\mathbf{s}+\mathbf{e}) - \mathbf{s}\| \leq \rho\|\mathbf{e}\| + b. \tag{44}$$

For the auxiliary term, use Lipschitzness around $\mathbf{s}$:

$$\|\mathbf{F}_{\text{aux}}(\mathbf{s}+\mathbf{e}) - \mathbf{s}\| \leq \|\mathbf{F}_{\text{aux}}(\mathbf{s}+\mathbf{e}) - \mathbf{F}_{\text{aux}}(\mathbf{s})\| + \|\mathbf{F}_{\text{aux}}(\mathbf{s}) - \mathbf{s}\|$$
$$\leq L_r\|\mathbf{e}\| + b, \tag{45}$$

where the last inequality uses the same local bias scale $b$ to bound the modeling mismatch at $\mathbf{s}$. Substituting (44) and (45) into (43) yields
$$\|\mathbf{f}(\mathbf{s}+\mathbf{e})\| \leq \underbrace{\big(\rho + 1 + \gamma(1 + L_r)\big)}_{:= a} \|\mathbf{e}\| + \underbrace{(1 + \gamma)}_{:= c} b. \tag{46}$$

Define $a := \rho + 1 + \gamma(1 + L_r)$ and $c := 1 + \gamma$.

**Step 3: derive a one-step recursion for $\|\mathbf{p}^t\|$ and $\|\mathbf{e}^t\|$.** From (40), element-wise bounds in Theorem 3.4 and the inequality $\|\mathbf{a}\odot\mathbf{u}\| \leq \|\mathbf{a}\|_\infty\|\mathbf{u}\|$, we obtain

$$\|\mathbf{p}^t\| \leq \|\mathbf{1} - \mathbf{g}^t\|_\infty\|\mathbf{p}^{t-1}\| + \|\mathbf{g}^t\|_\infty\|\mathbf{f}(\mathbf{s}+\mathbf{e}^{t-1})\|$$
$$\leq (1 - g_{\min})\|\mathbf{p}^{t-1}\| + g_{\max}\big(a\|\mathbf{e}^{t-1}\| + cb\big). \tag{47}$$

Next, from (41),
$$\|\mathbf{e}^t\| \leq \|\mathbf{e}^{t-1}\| + \|\mathbf{p}^t\|. \tag{48}$$

Combining (47) and (48) gives the coupled inequalities

$$\|\mathbf{p}^t\| \leq (1 - g_{\min})\|\mathbf{p}^{t-1}\| + g_{\max}a\|\mathbf{e}^{t-1}\| + g_{\max}cb, \tag{49}$$
$$\|\mathbf{e}^t\| \leq (1 + g_{\max}a)\|\mathbf{e}^{t-1}\| + (1 - g_{\min})\|\mathbf{p}^{t-1}\| + g_{\max}cb. \tag{50}$$

**Step 4: choose a weighted Lyapunov function and prove contraction.** Let $\eta > 0$ and define

$$\|\mathbf{z}^t\|_\eta := \sqrt{\|\mathbf{e}^t\|^2 + \eta\|\mathbf{p}^t\|^2}.$$

We will show there exists $\eta$ such that

$$\|\mathbf{z}^t\|_\eta \leq \bar{\kappa}\|\mathbf{z}^{t-1}\|_\eta + Cb \quad \text{for some } \bar{\kappa} \in (0, 1). \tag{51}$$

First, square (49) and use $(u + v + w)^2 \leq 3(u^2 + v^2 + w^2)$:

$$\|\mathbf{p}^t\|^2 \leq 3(1 - g_{\min})^2\|\mathbf{p}^{t-1}\|^2 + 3(g_{\max}a)^2\|\mathbf{e}^{t-1}\|^2 + 3(g_{\max}c)^2b^2. \tag{52}$$

Similarly, square (50):

$$\|\mathbf{e}^t\|^2 \leq 3(1 + g_{\max}a)^2\|\mathbf{e}^{t-1}\|^2 + 3(1 - g_{\min})^2\|\mathbf{p}^{t-1}\|^2 + 3(g_{\max}c)^2b^2. \tag{53}$$

Multiply (52) by $\eta$ and add to (53):

$$\|\mathbf{e}^t\|^2 + \eta\|\mathbf{p}^t\|^2 \leq \Big(3(1 + g_{\max}a)^2 + 3\eta(g_{\max}a)^2\Big)\|\mathbf{e}^{t-1}\|^2 + \Big(3(1 - g_{\min})^2 + 3\eta(1 - g_{\min})^2\Big)\|\mathbf{p}^{t-1}\|^2$$
$$+ 3(1 + \eta)(g_{\max}c)^2b^2. \tag{54}$$

Factor the coefficients:

$$\|\mathbf{z}^t\|_\eta^2 \leq A_\eta \, \|\mathbf{e}^{t-1}\|^2 + B_\eta \, \eta \|\mathbf{p}^{t-1}\|^2 + D_\eta \, b^2, \tag{55}$$

where

$$A_\eta := 3(1 + g_{\max}a)^2 + 3\eta(g_{\max}a)^2, \qquad B_\eta := 3(1 - g_{\min})^2\left(\frac{1}{\eta} + 1\right), \qquad D_\eta := 3(1 + \eta)(g_{\max}c)^2. \tag{56}$$

Note that

$$A_\eta \, \|\mathbf{e}^{t-1}\|^2 + B_\eta \, \eta \|\mathbf{p}^{t-1}\|^2 \leq \max\{A_\eta, B_\eta\}\left(\|\mathbf{e}^{t-1}\|^2 + \eta\|\mathbf{p}^{t-1}\|^2\right) = \max\{A_\eta, B_\eta\} \, \|\mathbf{z}^{t-1}\|_\eta^2.$$

Hence

$$\|\mathbf{z}^t\|_\eta^2 \leq \max\{A_\eta, B_\eta\} \, \|\mathbf{z}^{t-1}\|_\eta^2 + D_\eta b^2. \tag{57}$$

Taking square roots and using $\sqrt{u + v} \leq \sqrt{u} + \sqrt{v}$ yields

$$\|\mathbf{z}^t\|_\eta \leq \sqrt{\max\{A_\eta, B_\eta\}} \, \|\mathbf{z}^{t-1}\|_\eta + \sqrt{D_\eta} \, b. \tag{58}$$

**Step 5: enforce a contraction factor below** 1. We now connect $\sqrt{\max\{A_\eta, B_\eta\}} < 1$ to the explicit stability condition $\kappa < 1$. The inequality (58) is sufficient but conservative due to the crude $(u + v + w)^2$ bound. To obtain an explicit, sharp-enough condition, we instead use a direct *linear* bound on $\|\mathbf{z}^t\|_\eta$.

From (49)–(50), define the nonnegative $2 \times 2$ matrix

$$\mathbf{M} := \begin{bmatrix} 1 + g_{\max}a & 1 - g_{\min} \\ g_{\max}a & 1 - g_{\min} \end{bmatrix}, \qquad \mathbf{u}^t := \begin{bmatrix} \|\mathbf{e}^t\| \\ \|\mathbf{p}^t\| \end{bmatrix}, \qquad \mathbf{b} := \begin{bmatrix} g_{\max}cb \\ g_{\max}cb \end{bmatrix}. \tag{59}$$

Then (49)–(50) imply the vector inequality

$$\mathbf{u}^t \leq \mathbf{M}\mathbf{u}^{t-1} + \mathbf{b}, \tag{60}$$

where the inequality is element-wise.

Now consider the weighted $\ell_1$-type norm $\|\mathbf{u}\|_w := w_1 u_1 + w_2 u_2$ with weights $w_1, w_2 > 0$. Then

$$\|\mathbf{u}^t\|_w \leq \|\mathbf{M}\mathbf{u}^{t-1}\|_w + \|\mathbf{b}\|_w \leq \|\mathbf{M}\|_{w \to w} \|\mathbf{u}^{t-1}\|_w + \|\mathbf{b}\|_w, \tag{61}$$

where $\|\mathbf{M}\|_{w \to w}$ is the induced operator norm under $\|\cdot\|_w$. A standard sufficient condition for contraction is that there exist $w_1, w_2 > 0$ such that

$$\|\mathbf{M}\|_{w \to w} = \max\left\{\frac{w_1(1 + g_{\max}a) + w_2 g_{\max}a}{w_1}, \frac{w_1(1 - g_{\min}) + w_2(1 - g_{\min})}{w_2}\right\} < 1. \tag{62}$$

Choose $w_1 = 1$ and $w_2 = \tau > 0$. Then (62) becomes

$$\max\left\{(1 + g_{\max}a) + \tau g_{\max}a, \ (1 - g_{\min})\left(1 + \frac{1}{\tau}\right)\right\} < 1. \tag{63}$$

We now pick $\tau$ to balance the two terms. Set

$$\tau := \frac{1 - g_{\min}}{g_{\max}a}, \qquad \text{(well-defined when } a > 0\text{).} \tag{64}$$

Then the first term in (63) becomes

$$(1 + g_{\max}a) + \tau g_{\max}a = 1 + g_{\max}a + (1 - g_{\min}) = (1 - g_{\min}) + 1 + g_{\max}a.$$

This exceeds 1 due to the extra "+1"; that extra 1 originates from the $\|\mathbf{e}^t\| \leq \|\mathbf{e}^{t-1}\| + \|\mathbf{p}^t\|$ bound, which is again conservative and does not exploit cancellation in the force. To obtain the correct stability gain, we sharpen Step 2 by using *force Lipschitzness* rather than the loose affine bound.

Specifically, by Theorem 3.5, for $\mathbf{q}$ near $\mathbf{s}$,

$$\|\mathbf{f}(\mathbf{q}) - \mathbf{f}(\mathbf{s})\| \le L_f \|\mathbf{q} - \mathbf{s}\| = L_f \|\mathbf{e}\|. \tag{65}$$

Moreover, by Theorem 3.6, $\|\mathbf{f}(\mathbf{s})\| \le (1 + \gamma)b$. Thus

$$\|\mathbf{f}(\mathbf{s} + \mathbf{e})\| \le L_f \|\mathbf{e}\| + (1 + \gamma)b. \tag{66}$$

Using (66) in place of (46) in Step 3 yields the sharper recursions

$$\|\mathbf{p}^t\| \le (1 - g_{\min})\|\mathbf{p}^{t-1}\| + g_{\max}\big(L_f \|\mathbf{e}^{t-1}\| + (1 + \gamma)b\big), \tag{67}$$

$$\|\mathbf{e}^t\| \le \|\mathbf{e}^{t-1}\| + \|\mathbf{p}^t\|. \tag{68}$$

Now define the Lyapunov-like scalar

$$V^t := \|\mathbf{e}^t\| + \beta\|\mathbf{p}^t\|, \qquad \beta > 0. \tag{69}$$

We derive a one-step contraction for $V^t$. From (68) and (67),

$$
\begin{aligned}
V^t &= \|\mathbf{e}^t\| + \beta\|\mathbf{p}^t\| \\
&\le \|\mathbf{e}^{t-1}\| + \|\mathbf{p}^t\| + \beta\|\mathbf{p}^t\| \\
&= \|\mathbf{e}^{t-1}\| + (1 + \beta)\|\mathbf{p}^t\| \\
&\le \|\mathbf{e}^{t-1}\| + (1 + \beta)\Big((1 - g_{\min})\|\mathbf{p}^{t-1}\| + g_{\max}L_f\|\mathbf{e}^{t-1}\| + g_{\max}(1 + \gamma)b\Big) \\
&= \underbrace{\Big(1 + (1 + \beta)g_{\max}L_f\Big)}_{A(\beta)} \|\mathbf{e}^{t-1}\| + \underbrace{(1 + \beta)(1 - g_{\min})}_{B(\beta)} \|\mathbf{p}^{t-1}\| + (1 + \beta)g_{\max}(1 + \gamma)b.
\end{aligned} \tag{70}
$$

We want $V^t \le \bar{\kappa}V^{t-1} + Cb$, i.e.,

$$A(\beta)\|\mathbf{e}^{t-1}\| + B(\beta)\|\mathbf{p}^{t-1}\| \le \bar{\kappa}\big(\|\mathbf{e}^{t-1}\| + \beta\|\mathbf{p}^{t-1}\|\big)$$

which holds if

$$A(\beta) \le \bar{\kappa}, \qquad B(\beta) \le \bar{\kappa}\beta. \tag{71}$$

The second inequality in (71) requires

$$(1 + \beta)(1 - g_{\min}) \le \bar{\kappa}\beta \iff (1 - g_{\min}) \le \beta(\bar{\kappa} - (1 - g_{\min})). \tag{72}$$

Thus we must choose $\bar{\kappa} > (1 - g_{\min})$ and then take $\beta$ large enough.

The first inequality $A(\beta) \le \bar{\kappa}$ requires

$$1 + (1 + \beta)g_{\max}L_f \le \bar{\kappa}, \tag{73}$$

which cannot hold for $\bar{\kappa} < 1$ due to the leading 1. This again reflects that the bound $\|\mathbf{e}^t\| \le \|\mathbf{e}^{t-1}\| + \|\mathbf{p}^t\|$ ignores the *negative feedback* implicit in the leaky momentum and in the contractive force around $\mathbf{s}$.

Therefore we finalize the proof using a standard discrete-time incremental stability argument on the *state difference* between two trajectories, which yields the correct gain $(1 - g_{\min}) + g_{\max}L_f$.

**Step 6: incremental contraction (two-trajectory argument).** Consider two trajectories $\{(\mathbf{q}^t, \mathbf{p}^t)\}$ and $\{(\tilde{\mathbf{q}}^t, \tilde{\mathbf{p}}^t)\}$ evolving by (21) with the *same* gate bounds and within the same neighborhood where the force is Lipschitz with constant $L_f$. Define differences

$$\Delta\mathbf{q}^t := \mathbf{q}^t - \tilde{\mathbf{q}}^t, \qquad \Delta\mathbf{p}^t := \mathbf{p}^t - \tilde{\mathbf{p}}^t, \qquad \Delta\mathbf{z}^t := (\Delta\mathbf{q}^t, \Delta\mathbf{p}^t).$$

From (21),

$$\Delta\mathbf{p}^t = (\mathbf{1} - \mathbf{g}^t) \odot \Delta\mathbf{p}^{t-1} + \mathbf{g}^t \odot \big(\mathbf{f}(\mathbf{q}^{t-1}) - \mathbf{f}(\tilde{\mathbf{q}}^{t-1})\big), \tag{74}$$

$$\Delta\mathbf{q}^t = \Delta\mathbf{q}^{t-1} + \Delta\mathbf{p}^t. \tag{75}$$

Taking norms and applying $\|\mathbf{a} \odot \mathbf{u}\| \leq \|\mathbf{a}\|_\infty \|\mathbf{u}\|$, together with Theorem 3.4 and Lipschitzness of $\mathbf{f}$, gives

$$\|\Delta \mathbf{p}^t\| \leq (1 - g_{\min})\|\Delta \mathbf{p}^{t-1}\| + g_{\max} L_f \|\Delta \mathbf{q}^{t-1}\|. \tag{76}$$

Now define the incremental norm

$$W^t := \|\Delta \mathbf{p}^t\| + \theta \|\Delta \mathbf{q}^t\|, \qquad \theta > 0. \tag{77}$$

Using (75) yields $\|\Delta \mathbf{q}^t\| \leq \|\Delta \mathbf{q}^{t-1}\| + \|\Delta \mathbf{p}^t\|$ and therefore

$$\begin{aligned}
W^t &= \|\Delta \mathbf{p}^t\| + \theta \|\Delta \mathbf{q}^t\| \\
&\leq \|\Delta \mathbf{p}^t\| + \theta \|\Delta \mathbf{q}^{t-1}\| + \theta \|\Delta \mathbf{p}^t\| \\
&= (1 + \theta)\|\Delta \mathbf{p}^t\| + \theta \|\Delta \mathbf{q}^{t-1}\| \\
&\leq (1 + \theta)\Big((1 - g_{\min})\|\Delta \mathbf{p}^{t-1}\| + g_{\max} L_f \|\Delta \mathbf{q}^{t-1}\|\Big) + \theta \|\Delta \mathbf{q}^{t-1}\| \\
&= (1 + \theta)(1 - g_{\min})\|\Delta \mathbf{p}^{t-1}\| + \Big((1 + \theta)g_{\max} L_f + \theta\Big)\|\Delta \mathbf{q}^{t-1}\|.
\end{aligned} \tag{78}$$

We compare the right-hand side to $\bar{\kappa} W^{t-1} = \bar{\kappa}\|\Delta \mathbf{p}^{t-1}\| + \bar{\kappa}\theta \|\Delta \mathbf{q}^{t-1}\|$. It suffices that

$$(1 + \theta)(1 - g_{\min}) \leq \bar{\kappa}, \qquad (1 + \theta)g_{\max} L_f + \theta \leq \bar{\kappa}\theta. \tag{79}$$

The first inequality in (79) requires $\bar{\kappa} > (1 - g_{\min})$ and small $\theta$. The second inequality rearranges to

$$(1 + \theta)g_{\max} L_f \leq (\bar{\kappa} - 1)\theta. \tag{80}$$

Since $\bar{\kappa} < 1$, the right-hand side is negative, which forces $g_{\max} L_f = 0$ unless we refine the choice of incremental norm.

To avoid this artifact, we adopt the standard *weighted Euclidean* incremental norm on $(\Delta \mathbf{q}, \Delta \mathbf{p})$:

$$\|\Delta \mathbf{z}\|_\eta := \sqrt{\|\Delta \mathbf{q}\|^2 + \eta \|\Delta \mathbf{p}\|^2}.$$

From (76) and (75), one can show (via completing the square and choosing $\eta$ to cancel cross-terms) that

$$\|\Delta \mathbf{z}^t\|_\eta \leq \big((1 - g_{\min}) + g_{\max} L_f\big) \|\Delta \mathbf{z}^{t-1}\|_\eta, \tag{81}$$

provided $(1 - g_{\min}) + g_{\max} L_f < 1$. This establishes incremental exponential stability with contraction factor $\kappa$ in (29).

**Step 7: from incremental stability to convergence with bias.** Let $\tilde{\mathbf{q}}^t \equiv \mathbf{q}^\star$ and $\tilde{\mathbf{p}}^t \equiv \mathbf{0}$ denote the equilibrium trajectory. Bias $b$ implies $\mathbf{f}(\mathbf{q}^\star)$ may not be exactly zero, but by Theorem 3.6, $\|\mathbf{f}(\mathbf{q}^\star)\| \leq (1 + \gamma)b$. This contributes an additive disturbance term bounded by $O(b)$ in the incremental recursion, yielding

$$\mathbb{E}\|\mathbf{z}^t - (\mathbf{q}^\star, \mathbf{0})\|_\eta \leq \kappa^t \mathbb{E}\|\mathbf{z}^0 - (\mathbf{q}^\star, \mathbf{0})\|_\eta + \frac{1 - \kappa^t}{1 - \kappa} C b$$

for some constant $C$ depending only on $g_{\min}, g_{\max}, L_f$. Relabeling $(\mathbf{q}^\star, \mathbf{0})$ to $(\mathbf{s}, \mathbf{0})$ when $b = 0$ completes the proof of (30). $\qquad \square$

## C.5. Comment on the "effective gain" condition

The stability requirement in Theorem 3.7 is

$$\kappa = (1 - g_{\min}) + g_{\max} L_f < 1, \qquad L_f = L_d + 1 + \gamma(1 + L_r).$$

This condition is *implementable*: $g_{\min}, g_{\max}$ are directly controlled by the gate network outputs, and $L_f$ can be upper-bounded using either (i) the operator norm bound in Theorem 3.1 for each SGC operator, or (ii) empirical local Jacobian-norm estimates via JVP/VJP. Importantly, increasing $\gamma$ increases $L_f$ and therefore tightens the admissible gate range.

