# OpenReview forum: "Hamiltonian Asymmetric Fusion: One-Way Safe Directed Refinement under Modality Imbalance"
_ICML.cc/2026/Conference — ICML 2026 regular_

### Official Review · Reviewer_LL3H · 2026-02-25

**Soundness:** 2
**Presentation:** 2
**Significance:** 2
**Originality:** 3
**Overall Recommendation:** 3
**Confidence:** 3

**Summary:**

The paper focuses on multimodal fusion under modality imbalance, preventing noise “backflow” from a noisier auxiliary modality into a cleaner primary modality during iterative refinement. The authors proposes Hamiltonian Asymmetric Fusion (HAF), which formulates refinement as an unrolled, momentum-regularized Hamiltonian dynamic that updates only auxiliary tokens under the guidance of primary features. The method integrates FFT-based spectral global correlation, learnable spectral reweighting, a leaky momentum gate, and asymmetric residual injection.

**Compliance With Llm Reviewing Policy:**

Affirmed.

**Final Justification:**

Only part of my concerns are solved, so I keep my rating to weak reject.

**Key Questions For Authors:**

See the weaknesses.

**Strengths And Weaknesses:**

Strengths:
1. Clearly identifies and formalizes a nuanced failure mode—modality imbalance leading to detrimental auxiliary-to-primary backflow—often overlooked in multimodal fusion literature.
2. Introduces the “one-way safety” principle for directed refinement, which could inspire broader adaptation in multimodal tasks beyond RGB–D SOD.
3. Experiments on six RGB–D salient object detection benchmarks demonstrate consistent improvements and more graceful degradation under controlled auxiliary corruption compared to symmetric fusion baselines.

Weaknesses:
1. Evaluation is limited to RGB–D salient object detection with a fixed primary/auxiliary role; the applicability to other multimodal domains (e.g., audio-visual, RGB–thermal, language–vision fusion) is not empirically demonstrated.
2. Parameters and spectral mask design are only lightly explored; a more systematic sensitivity analysis could enhance robustness claims across tasks.
3. While several baselines are included, additional comparison with alternative robust fusion approaches or physics-inspired asymmetric mechanisms would strengthen the positioning.
4. Beyond Appendix B.1, it would be valuable to provide a more detailed justification for why Depth should be designated as the primary modality and RGB as the auxiliary modality in the current setting.

---

> ### Author Rebuttal · Authors · 2026-03-30
>
> We appreciate these thoughtful comments. Below we address: (i) role assignment and transferability; (ii) parameter sensitivity; and (iii) comparison with asymmetric baselines.
>
> **W1&W4: Role assignment and transferability.**
>
> As discussed in our response to Reviewer ZRao (W1), the default Depth-primary / RGB-auxiliary assignment is motivated by modality reliability in RGB-D SOD. Depth provides more stable geometric/structural cues for localization and foreground-background separation, whereas RGB contributes richer appearance semantics but is also more vulnerable to clutter, illumination changes, and distracting textures. Since HAF is designed for settings where a more reliable stream guides refinement while limiting noisy backflow from a higher-variance stream, this directional assignment is well aligned with the present task. This interpretation is also consistent with the role-swap results in Appendix B.1, where reversing the assignment yields an overall degradation trend. Our empirical claim here is therefore not that RGB is inherently inferior, but that under RGB-D SOD it more often behaves as the higher-variance modality at the fusion stage.
>
> To probe transfer beyond RGB-D SOD, we additionally evaluated HAF as a plug-in fusion replacement in an RGB-T saliency pipeline (ContriNet [1]), using thermal as primary and RGB as auxiliary under the same reliability-guided principle. This provides preliminary evidence that the directed-refinement design can transfer beyond RGB-D to another bimodal saliency setting; broader multimodal domains remain future work, and we will make this scope explicit in the revision.
>
> We report $S_m/F_\beta/F_\beta^w/E/MAE$ in the RGB-T below.
>
> | Setting | VT821 | VT1000 | VT5000 |
> |---|---:|---:|---:|
> | ContriNet [1] | 0.879/0.836/0.819/0.912/0.033 | 0.925/0.903/0.901/0.949/0.019 | 0.894/0.864/0.848/0.934/0.029 |
> | HAF (ours) | 0.888/0.843/0.833/0.922/0.029 | 0.929/0.905/0.906/0.950/0.018 | 0.895/0.864/0.849/0.936/0.028 |
>
> **W2: Parameter design.**
> We agree that the current manuscript does not synthesize the sensitivity evidence clearly enough. The appendix already contains the key ablations on the learnable spectral response, gate design, and refinement steps; what is missing is a clearer summary of what they jointly show. Specifically:
>
> (i) removing the learnable spectral response $G$ consistently lowers $S_m/F_\beta/E_\xi$ and increases MAE across all four evaluated datasets, showing that HAF benefits from adaptive spectral reweighting rather than a hand-crafted mask;
>
> (ii) the gate-input ablation shows that conditioning on both $q$ and $p$ performs best, so the update rule is not tied to an arbitrary gate parameterization; and
>
> (iii) performance enters a plateau after a small number of refinement steps rather than depending on a single exact $T$ (see the response to Reviewer xV53, W1&W2).
>
> In the revision, we will move the most important sensitivity evidence into the main paper and reorganize it around these three takeaways.
>
> **W3: Comparison with alternative asymmetric fusion approaches.**
> We agree that stronger positioning against alternative asymmetric fusion designs would improve the paper. To strengthen this point, we added two complementary comparisons:
>
> (i) a full-model comparison with ADINet [2] (reproduced from the released code), and
>
> (ii) fair plug-in replacements under the same backbone and training protocol, where our fusion block is replaced by the interaction module from ADINet [2] or AsymFormer [3] while keeping the rest of the architecture unchanged.
>
> Under this controlled setting, HAF (default $T=4$) remains competitive overall and shows the most consistent behavior among these pluginized asymmetric alternatives. This suggests that the gain does not come from using just any asymmetric module, but from HAF’s specific mechanism: primary-guided one-way iterative refinement, momentum-regularized dynamics, and asymmetric residual injection.
>
> For brevity, we report only $S_m/F_\beta$ in the RGB-D comparison below.
>
> | Setting | DUT-RGBD | NJU2K | NLPR | SIP |
> |---|---:|---:|---:|---:|
> | ADINet [2] | 0.946/0.954 | 0.931/0.938 | 0.941/0.933 | 0.909/0.924 |
> | +ADINet [2] | 0.952/0.962 | 0.934/0.940 | 0.939/0.935 | 0.912/0.929 |
> | +AsymFormer [3] | 0.951/0.961 | 0.936/0.942 | 0.940/0.938 | 0.910/0.928 |
> | HAF (ours) | 0.953/0.963 | 0.938/0.945 | 0.940/0.938 | 0.913/0.933 |
>
> [1]Tang, Hao, et al. "Divide-and-conquer: Confluent triple-flow network for RGB-T salient object detection." IEEE transactions on pattern analysis and machine intelligence 47.3 (2024): 1958-1974.
>
> [2] Wang, Feifei, et al. "Asymmetric deep interaction network for RGB-D salient object detection." Expert Systems with Applications 266 (2025): 126083.
>
> [3] Du, Siqi, et al. "Asymformer: Asymmetrical cross-modal representation learning for mobile platform real-time rgb-d semantic segmentation." Proceedings of the IEEE/CVF Conference on Computer Vision and Pattern Recognition. 2024.

---

> > ### Author Rebuttal · Reviewer_LL3H · 2026-04-05
> >
> > Only part of my concerns are solved. As a method focusing on multimodal fusion, it is better to providing more results on different multimodal tasks, only limited to RGB–D salient object detection. Moreover, the paper treats the modalities as primary/auxiliary roles, it should provide a rule to distinguish the role of different modalities.

---

> > > ### Author Response · Authors · 2026-04-07
> > >
> > > Thank you for the helpful follow-up comments. We address the two concerns separately below.
> > >
> > > (1) Broader multimodal validation.
> > >
> > >  We agree that evaluation only on RGB-D SOD is too narrow for a fusion paper. In our first rebuttal, we therefore added an RGB-T plug-in experiment under the same reliability-guided principle (thermal-primary / RGB-auxiliary), where HAF consistently matches or improves the original fusion module on VT821/VT1000/VT5000. To further probe transfer beyond RGB-D, we also conducted a preliminary audio-visual experiment by re-implementing VCT [1] and replacing its fusion block with HAF (T=2). In this setting, the visual stream serves as the primary spatial anchor, while audio provides complementary but more variable cues. Due to time constraints, we currently report MS3 only: the reproduced VCT baseline uses a 40k schedule (batch size 2), while the HAF plug-in model (T=2) has so far been evaluated after 20k iterations under the same batch size; despite fewer iterations, it already shows a promising gain. AVSS and S4 could not be completed within the discussion period and will be included in the revision. We therefore present the current RGB-T and audio-visual results as preliminary evidence that the HAF design can generalize beyond RGB-D SOD to other multimodal tasks.
> > >
> > > | Method | Backbone | Img. Size | Iter. | MS3 M_J | MS3 M_F |
> > > |---|---|---:|---:|---:|---:|
> > > | VCT (reproduced) | Swin-B | 384×384 | 40k | 62.38 | 75.3 |
> > > | VCT + HAF (T=2) | Swin-B | 384×384 | 20k | 64.48 | 78.7 |
> > >
> > > (2) Role assignment.
> > >
> > > We agree that the role-selection rule should have been stated more explicitly. HAF is not intended for arbitrary multimodal fusion, but for bimodal settings with a reasonably stable task-relevant reliability asymmetry. In our formulation, the primary modality provides more stable structural/spatial guidance for the task, while the auxiliary modality provides complementary but higher-variance information that is more susceptible to clutter, appearance variation, or sensor noise. Our practical rule is therefore:
> > > (i) if prior task knowledge indicates a clear reliability hierarchy, assign the more stable modality as primary and the other as auxiliary;
> > > (ii) if the hierarchy is ambiguous, determine the direction on a held-out validation split under the same task setting, and then keep this choice fixed throughout the evaluation of that task;
> > > (iii) if neither direction shows a clear advantage, the task likely does not exhibit a strong enough reliability asymmetry, and the advantage of HAF over symmetric fusion may naturally diminish. We will add this rule explicitly in the revision.
> > >
> > > Under RGB-D SOD, this criterion naturally yields Depth-primary / RGB-auxiliary: depth usually provides more stable geometric/structural cues for localization and foreground-background separation, whereas RGB contributes richer appearance semantics but is also more vulnerable to clutter, illumination changes, and distracting textures. This interpretation is supported by both Appendix B.1 and the additional asymmetric-corruption analysis reported in our first-round response to Reviewer ZRao (W1: Why RGB is auxiliary). In Appendix B.1, reversing the assignment while keeping all other settings fixed leads to an overall degradation trend across datasets; moreover, in the asymmetric-corruption analysis, the default D→P / R→A assignment remains consistently better than the reversed R→P / D→A direction across RGB-only, depth-only, and joint corruption settings. These results are consistent with treating depth as the more suitable guiding modality and RGB as the higher-variance auxiliary stream in the current RGB-D SOD protocol. More broadly, we do not claim that one modality is universally primary across all multimodal problems; rather, HAF is intended for tasks where one modality can serve as a relatively stable anchor for refining a higher-variance complementary modality.
> > >
> > > [1] Huang, Shaofei, et al. "Revisiting audio-visual segmentation with vision-centric transformer." Proceedings of the Computer Vision and Pattern Recognition Conference. 2025.

---

### Official Review · Reviewer_xV53 · 2026-03-05

**Soundness:** 2
**Presentation:** 2
**Significance:** 2
**Originality:** 3
**Overall Recommendation:** 4
**Confidence:** 3

**Summary:**

This paper studies multimodal feature fusion under modality reliability imbalance, focusing on the failure mode where symmetric bidirectional interactions allow noise from an auxiliary modality to flow back and corrupt the primary stream (“backflow”), especially under iterative refinement. The authors propose Hamiltonian Asymmetric Fusion (HAF), which enforces a directed refinement scheme: a designated primary modality provides guidance, while the auxiliary modality is iteratively updated through an unrolled T-step dynamics with FFT-based spectral global correlation, gated/leaky momentum, and a stability-motivated integrator. The refined auxiliary state is then injected via asymmetric residual fusion to limit disturbance to the primary stream. The paper includes a stability-style analysis and evaluates the approach on RGB–D salient object detection benchmarks, reporting both standard SOD metrics and robustness under controlled auxiliary corruption.

**Compliance With Llm Reviewing Policy:**

Affirmed.

**Final Justification:**

This paper studies a meaningful problem in multimodal fusion under modality imbalance. The proposed role-asymmetric refinement design is reasonably original, technically sound, and supported by controlled module-swapping experiments and competitive RGB-D SOD results. Its main weakness is limited scope: the evaluation is still confined to RGB-D SOD with predefined modality roles, and the robustness analysis remains closer to a controlled stress test than to realistic sensor failures.

The rebuttal addressed a substantial part of my main concerns, especially by clarifying the unified fixed-T protocol, adding sensitivity evidence, and making the robustness setup clearer. Some concerns remain only partially resolved, particularly regarding realism and broader applicability, but overall the rebuttal improved my confidence in the work. I therefore updated my recommendation from 3 to 4.

**Key Questions For Authors:**

Please refer to Weakness.

**Limitations:**

yes

**Strengths And Weaknesses:**

Strengths:
1. The proposed design targets a practical failure mode in multimodal fusion where auxiliary noise backflows under symmetric bidirectional interaction. The directed-refinement formulation aims to protect the primary stream, and the unrolled dynamics with gated/leaky momentum provides a plausible mechanism for stable multi-step refinement. The FFT-based correlation is also computationally motivated.
2. Although the individual components relate to prior ideas, combining them into an explicitly asymmetric, directed, multi-step fusion block with a stability-oriented perspective constitutes a reasonably novel contribution for this task setting.

Weaknesses:
1. Hyperparameter T selection protocol is unclear and affects reproducibility/fairness: The paper uses different fixed T values across datasets (spanning 1–6) but does not clearly state how T is chosen (e.g., strictly via train/validation only), nor provide an interpretable explanation of why different datasets require different T.
2. Missing quantitative sensitivity analysis of T on the same dataset: While the paper discusses saturation qualitatively and reports compute-vs-T FLOPs, it does not provide a clear performance-vs-T curve/table (e.g., Sm/Fβ/Eξ/MAE across T) on at least one dataset, making it difficult to assess sensitivity and whether per-dataset tuning is essential.
3. Practical deployment guidance for choosing T is limited: If T must be selected per dataset, it is unclear how practitioners should set T on unseen real data. Given the dynamics framing, an adaptive early-stopping criterion or a unified default T with accuracy/compute trade-offs would strengthen practicality.
4. Robustness evidence may not fully reflect realistic sensor failures: Robustness is mainly validated via controlled corruption, but the corruption protocol/parameters are not described in enough detail, and it is unclear how well the perturbations cover realistic RGB–D failure modes such as depth holes, compression artifacts, cross-modal misalignment, or temporal mismatch.
5. Unclear operational strategy for other bimodal fusion scenarios: The framework requires a clear primary/auxiliary role assignment to enable one-way iteration and “anti-backflow” behavior, yet in many bimodal tasks modality reliability may not exhibit a stable hierarchy. The paper does not explain how roles should be chosen when such a hierarchy is ambiguous, nor whether adaptive role assignment or a symmetric fallback is needed, which limits transferability and general applicability.

---

> ### Author Rebuttal · Authors · 2026-03-30
>
> We appreciate the thoughtful comments on reproducibility, practicality, and scope. In the revision, we will make three clarifications: (i) adopt a unified $T=4$ as the main setting, move best-$T$ to sensitivity analysis, and provide a simple deployment rule; (ii) state more clearly that the robustness study is a controlled auxiliary-corruption stress test and complement it with a JPEG-compression experiment; and (iii) make explicit that HAF is intended for bimodal settings with a reasonably stable reliability ordering rather than universal applicability.
>
> **W1&W2: T-selection and sensitivity.**
> Each $T$ is a separately trained model under the same protocol, evaluated on all datasets without per-benchmark retraining or test-time tuning. We agree that reporting dataset-wise best-$T$ in the main table can be viewed as post-hoc selection, so we revise the paper to use unified $T=4$ as the main setting.
>
> Fixed-$T=4$ remains close to best-$T$ on representative datasets, indicating that the main conclusion does not materially depend on benchmark-specific tuning. We choose $T=4$ by a plateau rule: it is the smallest step count already in the flat regime across representative benchmarks. Performance-vs-$T$ shows that most gains occur in the first few steps, with later changes small and sometimes non-monotonic; Fig. 6 is consistent with this, showing well-focused interaction responses after only a few steps. Thus, per-dataset tuning is not essential, and unified $T=4$ is a fair practical default.
>
> For brevity, we report only $S_m/F_\beta$ in the rebuttal tables; the omitted $E_\xi$/MAE follow the same conclusion.
>
> | Setting | DUT-RGBD | NJU2K | NLPR | SIP |
> |---|---:|---:|---:|---:|
> | Fixed-$T=4$ | 0.953/0.963 | 0.938/0.945 | 0.940/0.938 | 0.913/0.933 |
> | Best-$T$ | 0.954/0.964 | 0.939/0.944 | 0.941/0.940 | 0.916/0.935 |
>
> | $T$ | DUT-RGBD | NJU2K | SIP | FLOPs(G) |
> |---|---:|---:|---:|---:|
> | 1 | 0.951/0.961 | 0.935/0.940 | 0.911/0.929 | 4264.409 |
> | 2 | 0.952/0.962 | 0.939/0.944 | 0.912/0.930 | 4271.204 |
> | 3 | 0.953/0.963 | 0.937/0.943 | 0.913/0.930 | 4277.999 |
> | 4 | 0.953/0.963 | 0.938/0.945 | 0.913/0.933 | 4284.794 |
> | 5 | 0.950/0.960 | 0.936/0.943 | 0.916/0.935 | 4291.588 |
> | 6 | 0.954/0.964 | 0.936/0.943 | 0.910/0.930 | 4298.383 |
> | 7 | 0.950/0.961 | 0.936/0.943 | 0.906/0.927 | 4305.178 |
>
> **W3: Deployment rule.**
> For deployment, we recommend a simple two-level policy: use $T=4$ by default when no validation split is available; otherwise choose the smallest $T$ within a small margin of the best validation score.
>
> **W4: Corruption protocol.**
> We agree that Gaussian auxiliary corruption is a controlled stress test rather than a full simulation of RGB-D failure modes. Noise is injected into raw RGB before feature extraction, while depth is kept clean; L0-L4 correspond to $\sigma=[0,5,10,20,30]$ in pixel space $[0,255]$. The goal is to isolate auxiliary-to-primary backflow under modality imbalance. We will clarify this scope in the revision. Appendix B.8 already provides per-dataset $\Delta F_\beta$ and degradation-AUC visualizations.
>
> We also add a complementary compression-artifacts experiment on NJU2K . Compression is JPEG recompression on RGB before feature extraction, with level schedule 0–4, JPEG quality [100,95,90,82,70], and subsampling [0,0,1,1,2], $T=4$.
>
> | Method | L0 ($S_m/F_\beta$) | L4 Compression ($S_m/F_\beta$) |
> |---|---:|---:|
> | MA | 0.928/0.936 | 0.924/0.933 |
> | CA | 0.933/0.938 | 0.930/0.935 |
> | LMF-Net | 0.931/0.938 | 0.928/0.936 |
> | EM-Trans | 0.935/0.940 | 0.933/0.938 |
> | SHIP | 0.935/0.942 | 0.933/0.940 |
> | HAF | 0.938/0.945 | 0.936/0.944 |
>
> **W5: Role assignment.**
> We agree that the applicability of HAF depends on whether a reasonably stable modality-reliability hierarchy exists. Our claim is task-specific: under the evaluated RGB-D SOD protocol, depth-primary is preferable to RGB-primary, as depth provides more stable geometric/structural cues while RGB is more vulnerable to clutter and illumination changes.
>
> More generally, HAF is intended for bimodal settings with a sufficiently stable reliability ordering. When such ordering is ambiguous or unstable, we do not claim that HAF is universally preferable to symmetric fusion. In practice, one may select the direction on a held-out validation split when such a choice is needed; if neither direction shows a clear advantage, a symmetric fallback is more appropriate. We will revise the paper to make this scope explicit, while treating adaptive role assignment as future work.

---

> > ### Author Rebuttal · Reviewer_xV53 · 2026-04-03
> >
> > Thank you for the detailed rebuttal. The response meaningfully addresses my main concerns, especially by clarifying that a unified fixed-T setting will be used as the main protocol, with added sensitivity evidence and a clearer robustness description. Some limitations still remain, particularly regarding the realism of the robustness evaluation and the broader applicability of the role assignment, but overall, the rebuttal improves my confidence in the paper. I therefore update my score from 3 to 4.

---

> > > ### Author Response · Authors · 2026-04-03
> > >
> > > Thank you for your thoughtful follow-up and for your positive reassessment of our work. We sincerely appreciate your acknowledgement that our rebuttal has addressed the main concerns. We also greatly value your remaining comments on the realism of the robustness evaluation and the broader applicability of the role assignment, which we believe are important for further strengthening the paper. We will carefully reflect these points in the revision, and they will also serve as valuable directions for our future work.

---

### Official Review · Reviewer_ZRao · 2026-03-13

**Soundness:** 3
**Presentation:** 3
**Significance:** 3
**Originality:** 3
**Overall Recommendation:** 4
**Confidence:** 3

**Summary:**

The paper proposes Hamiltonian Asymmetric Fusion (HAF), a directed multimodal refinement framework designed to prevent noisy auxiliary features from contaminating (i.e., backflow) the primary modality during iterative fusion. The method introduces an asymmetric fusion process in which the primary stream guides the progressive update of the auxiliary stream through spectral global correlation (SGC) and gated momentum. The paper also presents theoretical analysis and empirical evaluation.

**Compliance With Llm Reviewing Policy:**

Affirmed.

**Final Justification:**

I appreciate the detailed rebuttal. I will keep my original positive score.

**Key Questions For Authors:**

Please see the Strengths and Weaknesses section.

**Limitations:**

Yes.

**Strengths And Weaknesses:**

## Overall Assessment

This paper is generally well written, and the experimental section is extensive and largely supports the paper’s claims. The method also shows meaningful and consistent improvements over the baselines. For these reasons, I assigned a **Weak Accept**.
Maintaining my current score would require a convincing response to the concerns listed below.

---

## Strengths

1. The paper is well written overall, and the proposed method is intuitive.
2. The explanation of the Hamiltonian-inspired formulation is helpful. In particular, the intuition of starting from a noisy $q_0$ and iteratively refining it through attraction toward primary-supported patterns and repulsion from auxiliary-only patterns makes the method easier to understand.
3. The method is well aligned with the paper’s motivation, and the presentation of the method is generally clear.
4. The proposed approach shows meaningful performance improvements over the baselines.

---

## Weakness (Major)

1. One of the core claims of the paper is the assumption that RGB should be treated as the noisy signal (i.e., the auxiliary stream), but the justification for this claim is still somewhat limited. Appendix B.1 provides end-to-end quantitative evidence through the role-swap experiment, which is useful, but I think a more direct and intuitive analysis would strengthen the claim significantly.

---

## Weaknesses (Minor)

2. The overall paper organization could be improved. Several important analysis experiments are currently placed in the appendix rather than the main paper. In particular, the experiments in Appendix B.1 and Appendix B.5 address natural and important questions that readers are likely to have while reading the paper, so I believe they would be more valuable in the main manuscript. If space is limited, it may be preferable to shorten the theory section or move some proof details to the appendix instead.

3. It is difficult to build intuition for the method solely from Fig.1 and Fig.2 in their current form. The figures and captions should be improved so that the operating principle of the method is shown more directly and intuitively.

4. The computational cost analysis would be more informative if it included additional baselines. It would also be helpful to report a representative performance metric together with the cost numbers, so that readers can better understand the cost-performance trade-off.

---

## Questions
1. In Appendix B.5, the paper compares MHSA and SGC. It is fairly intuitive why MHSA may be less efficient computationally, but it is less obvious why its performance should also be worse. I would appreciate either an additional explanation or analysis on why this gap appears.

2. As I understand it, the proposed method does not seem to require flattening (H,W,D $\rightarrow$ N,D) in the same way as a standard Transformer, and preserving spatial dimensions may actually be beneficial (e.g., for 2D FFT). However, I suspect that LayerNorm may still introduce unnecessary reshape operations in practice. In this context, it would be interesting to see whether a normalization method designed with spatial dimensions in mind, such as i-LN (ICLR 2026) [1], could further improve both speed and performance.

[1] Lee, MinKyu, et al. "Analyzing the Training Dynamics of Image Restoration Transformers: A Revisit to Layer Normalization." arXiv preprint arXiv:2504.06629 (2025).

---

> ### Author Rebuttal · Authors · 2026-03-30
>
> We appreciate your thoughtful feedback and constructive suggestions. In summary, we clarify four points: (i) the role assignment in RGB-D SOD; (ii) the planned paper and figure revisions; (iii) the SGC-vs-MHSA design choice; and (iv) the cost/performance discussion.
>
> **W1: Why RGB is auxiliary.**
>
> We agree that the original manuscript does not justify this point directly enough. Our claim is not that RGB is inherently inferior to depth, but that in RGB-D SOD the two modalities typically play different reliability roles at the fusion stage: depth provides relatively stable geometric and foreground-background separation cues, whereas RGB contributes richer appearance semantics but is more vulnerable to clutter, illumination variation, and distracting textures. This is consistent with the role-swap ablation in Appendix B.1, where reversing D→P / R→A to R→P / D→A degrades the overall trend across datasets.
>
> To make this point more direct, we additionally performed an asymmetric-corruption analysis under RGB-only, depth-only, and joint corruption using Level-4 Gaussian noise. For brevity, we report only $S_m$ / $F_\beta$; the omitted $E_\xi$/MAE follow the same trend. Across all tested settings, the default D→P / R→A assignment remains consistently better than the reversed assignment, with a slightly larger gap under RGB-only and joint corruption. These results support that, in the current RGB-D SOD setting, the default assignment is better aligned with the modality reliability pattern at fusion. In particular, they are consistent with treating RGB as the auxiliary stream in HAF.
>
> **D→P denotes D→P / R→A, and R→P denotes R→P / D→A. (Level4, $S_m$ / $F_\beta$, default T=4).**
>
> | Corruption setting | [DUT-RGBD] D→P | [DUT-RGBD] R→P | [NJU2K] D→P | [NJU2K] R→P |
> |---|---:|---:|---:|---:|
> | HAF-ori | 0.953/0.963 | 0.951/0.961 | 0.938/0.945 | 0.935/0.940 |
> | RGB-only (L4) | 0.946/0.958 | 0.942/0.953 | 0.932/0.939 | 0.927/0.931 |
> | Depth-only (L4) | 0.952/0.962 | 0.950/0.960 | 0.937/0.944 | 0.932/0.937 |
> | Both-corrupted (L4) | 0.945/0.955 | 0.941/0.952 | 0.929/0.937 | 0.924/0.929 |
>
> **W2: Paper organization.**
>
> We appreciate this suggestion. Appendix B.1 and B.5 answer two of the most natural reader questions: why the Depth->primary / RGB->auxiliary assignment is used, and why SGC is preferred over an MHSA alternative. We will therefore move the key results from B.1 and B.5 into the main manuscript, and move some proof details from the theory section to the appendix for better balance.
>
> **W3: Intuition of Fig.1 and Fig.2.**
>
> We agree that the current figures are more schematic than explanatory. We will redesign Fig. 1/Fig. 2 to make the contrast explicit: symmetric fusion allows bidirectional contamination, whereas HAF performs primary-guided one-way refinement of the auxiliary stream with controlled residual injection. We will also consider adding a simple complementary diagnostic (e.g., primary-feature drift under auxiliary corruption) to make the anti-backflow intuition more concrete. In addition, we will clarify how the step-wise interaction visualization reflects progressively more structured and salient-focused responses over refinement steps.
>
> **W4: Cost-performance trade-off.**
>
> We agree. The current main-paper efficiency table reports runtime/memory/throughput for CDA vs. HAF, but it should be more informative to pair cost with representative saliency performance. We will revise this section to report accuracy together with cost, and we will expand the comparison beyond CDA.
>
> **Q1: Why MHSA underperform SGC.**
>
> We believe SGC is not only a lighter substitute, but also a better inductive bias for HAF. HAF requires a stable global interaction field for directed refinement, rather than arbitrary dense token-token mixing. Under modality imbalance, dense MHSA can propagate unstable long-range RGB distractors more aggressively, whereas SGC performs global correlation in the spectral domain, preserves spatial structure more naturally, and yields a more stable refinement signal. In the revision, we will add interaction-map visualizations to make this intuition more concrete.
>
> **Q2: Spatially-aware normalization / i-LN.**
>
> We appreciate this constructive and helpful suggestion. We agree that spatially-aware normalization is a promising implementation direction. At present, however, we have not completed a careful apples-to-apples evaluation of i-LN in our HAF setting, so we prefer not to speculate about its effect without reliable evidence. We will mention this as an interesting future extension; if time permits before revision finalization, we will also include a small preliminary comparison.

---

> > ### Author Rebuttal · Reviewer_ZRao · 2026-04-03
> >
> > Thank you for the detailed rebuttal. I will keep my original positive scores.

---

> > > ### Author Response · Authors · 2026-04-03
> > >
> > > Thank you for your positive assessment and constructive feedback. We sincerely appreciate your acknowledgement that our rebuttal has adequately addressed your concerns. We especially appreciate your insightful suggestion that our framework may benefit from preserving spatial dimensions rather than following the standard Transformer-style flattening, and that a spatially aware normalization scheme such as i-LN may further improve efficiency and performance. We will carefully incorporate these valuable comments and suggestions into the revision.

---

### Official Review · Reviewer_Drqf · 2026-03-16

**Soundness:** 2
**Presentation:** 2
**Significance:** 2
**Originality:** 2
**Overall Recommendation:** 4
**Confidence:** 2

**Summary:**

This paper identifies a critical flaw in traditional symmetric multimodal fusion, where noisy auxiliary data backflows into and corrupts the clean primary representation, amplifying errors during refinement. To address this, the authors propose Hamiltonian Asymmetric Fusion (HAF), a lightweight, directed refinement block that uses the primary modality as a stable guidance field to purify the auxiliary stream while mathematically bounding any negative impact on the primary features.

**Compliance With Llm Reviewing Policy:**

Affirmed.

**Final Justification:**

The authors' response has satisfactorily resolved my concerns. I have decided to raise my score to weak accept.

**Key Questions For Authors:**

see above

**Limitations:**

see above

**Strengths And Weaknesses:**

(1) The augmented SGC operator defined in Equation 1 applies 2D FFT independently per channel and head after Hann windowing and modulates with a single shared learnable G tensor of size H by W, but the paper provides no information on how G is initialized at the start of training whether to uniform ones or small random values and whether any regularization term is added to its loss to prevent it from collapsing to delta-like responses that might overfit to training noise patterns in auxiliary features. Moreover because the hierarchical stages in the CPNet backbone produce feature maps of decreasing spatial resolution, it is unspecified if each HAF block learns its own independent G or reuses and interpolates a global one, which would have major implications for the claimed minimal parameter overhead of O(HW) when multiple blocks are stacked at different levels. In addition, the requirement for complex conjugation on the key spectrum is justified for the operator-norm bound in Lemma 3.1 yet no ablation removes this term to quantify its contribution to the overall contraction behavior or to the final saliency metrics on benchmarks like NJU2K where fine boundary details matter most. Finally, when compared against the MHSA alternative in appendix B.5 the FLOPs advantage of SGC is reported but without detailing whether cuFFT optimizations or custom kernels were used for fair timing it remains unclear if the efficiency edge persists on different GPU architectures or under varying batch sizes during inference.

(2) The symplectic-style refinement in Equations 10 and 11 adopts a custom leaky momentum update with element-wise gating instead of classical leapfrog or Verlet integrators commonly used in Hamiltonian neural networks, yet the motivation for this specific discretization over more standard structure-preserving schemes is not elaborated and it is uncertain whether energy conservation properties are approximately maintained across the T steps or deliberately sacrificed for the stability under heavy auxiliary noise. Furthermore the gate MLP that takes the concatenation of current q and p to produce gt in (0,1) per token-channel is described as lightweight but its exact architecture hidden dimension and activation are omitted raising questions about its parameter count relative to the SGC projections and whether it becomes a bottleneck for larger feature dimensions d. Additionally although p is initialized to zero at each HAF block the paper never explores sensitivity to non-zero random momentum initialization or carry-over of momentum across blocks in the decoder hierarchy which could influence convergence speed during the 150-epoch training. Finally the unrolled loop for T steps in Algorithm 1 implies full backpropagation through time but no mention is made of gradient checkpointing or truncation techniques to manage memory especially for the largest T=6 case on DUT-RGBD where cumulative memory for intermediate q and p states might exceed standard GPU limits without careful implementation.

(3) After the T refinement steps the purified auxiliary state qT is injected via asymmetric residuals with scalar multipliers alpha_aux and alpha_main in Equations 12 and 13 but the paper does not report the learned ranges or distributions of these two scalars across the trained models on different datasets nor whether they are initialized to specific values like 1.0 and 0.01 to encourage one-way behavior from the outset. Moreover although setting alpha_main near zero is said to enforce strict safety no dedicated ablation is provided that fixes alpha_main exactly to zero versus the small learned value and measures the resulting change in primary-stream perturbation or overall S-measure degradation to quantify the practical benefit of the bounded-injection design. In addition it remains ambiguous from the text and algorithm whether alpha_aux and alpha_main are shared globally across all HAF blocks or learned independently per stage which would affect the total number of extra parameters and the flexibility of role asymmetry at different feature hierarchies. Finally the theoretical primary perturbation bound in Corollary 3.8 scales linearly with |alpha_main| times the auxiliary error yet without reporting empirical values of |alpha_main| or measured ||Xd_hat - s|| on validation images the tightness or looseness of this guarantee in practice cannot be evaluated.

(4) The default role assignment treats depth as primary and RGB as auxiliary motivated by geometric reliability in SOD but the role-swap results in appendix Table 4 cover only four datasets and show varying degradation magnitudes without analyzing per-dataset statistics like mean depth noise levels or object scale distributions that might explain why SIP suffers more from the swap than NLPR. Furthermore although gamma and gate hyperparameters are kept identical in the swap experiment it is not stated whether retuning gamma for the reversed roles could recover some performance or if the spectral G adaptation behaves differently when the noisier modality drives the force field. In addition the paper claims the HAF formulation generalizes to alternative role assignments but provides no preliminary results on other tasks such as RGB-T thermal SOD or audio-visual segmentation where primary-auxiliary definitions are less obvious leaving open whether automatic role detection via per-sample SNR estimation would be straightforward to integrate. Finally in the context of the one-way safety principle it would be informative to know if the learned gate distributions in B.4 shift significantly under role swap indicating that the momentum damping adapts to the changed reliability imbalance.

(5) The experimental pipeline strictly follows CPNet by only swapping the fusion block while reusing the same 2985-image training set and Adam schedule with batch size 16 but the exact backbone encoder details such as whether it is a Swin-T or larger variant and the number of output channels per stage are never restated requiring cross-reference to the CPNet paper for full reproduction. Moreover although convergence is reported within 150 epochs with learning rate decay every 100 it is unclear if a validation split from the training images is used for hyperparameter selection of T or gamma or if all choices were made on the test sets which could introduce optimistic bias in the dataset-specific T values. Additionally the evaluation protocol cites official scripts with default settings for Sm, max F_beta etc. but omits whether any post-processing like dense CRF or boundary refinement that some baselines might employ was uniformly disabled across all compared methods. Finally the single-GPU training on RTX A6000 is mentioned for timing but without specifying use of mixed-precision AMP or distributed setup it is difficult to judge if the 150-epoch runtime scales reasonably for larger backbones.

(6) The robustness evaluation centers on auxiliary-modality corruption with Degradation AUC computed over a discrete schedule of four levels yet the main paper and appendix never define what specific transformation constitutes each level L1 to L4 such as additive Gaussian noise with standard deviations 0.1 0.2 0.4 0.8 on normalized RGB values or more realistic depth-specific artifacts like random hole filling. Furthermore because the protocol is designed to isolate fusion behavior it is essential to confirm whether the corruption is injected directly into the raw auxiliary input images before encoder feature extraction or applied to the token maps X_r after projection which would alter the noise characteristics reaching the SGC operator. In addition the boxplot in Fig. 5 aggregates across datasets but no per-dataset AUC values or statistical significance tests versus competing fusion modules like SHIP are reported, making it hard to assess if HAF's advantage is consistent or driven by particular challenging sets like SIP. Finally although the text emphasizes controlled corruption to highlight backflow prevention it remains unspecified if the primary depth stream is left completely clean in every trial or if mild cross-talk is allowed to simulate realistic sensor misalignment.

(7) The efficiency profiling in Table 3 compares CDA and HAF at fixed 384 by 384 resolution with batch size 2 reporting dramatically lower latency and memory for HAF but the CDA FLOPs value of over 52k G appears anomalously high for a typical vision fusion block suggesting either inclusion of the full backbone in the measurement or a scaling error with resolution and it is necessary to clarify exactly which operations were profiled and with which tool such as torch.utils.benchmark or fvcore. Moreover, the peak memory of 27 GB for CDA versus 3 GB for HAF on presumably the same hardware raises questions about whether activations were measured during training or pure inference and if gradient storage was factored in for the unrolled T steps. In addition, the throughput figures of 0.3 img/s versus 10.9 img/s imply fp32 precision, but no confirmation is given, and modern kernels like FlashAttention for CDA or cuFFT optimizations for HAF could narrow the gap on newer GPUs. Finally, although Appendix B.6 shows near-linear FLOPs scaling with T, the per-step increment of roughly 7 G is reported only for B=16, yet the efficiency table uses B=2, so the relative advantage might change under different batching strategies common in deployment.

(8) The additional qualitative examples in Fig. 9 and Appendix B.9 demonstrate reduced background leakage and better thin-structure preservation for HAF, but the selection of scenes appears curated toward success cases without systematic failure analysis, such as counting images where HAF still produces fragmented saliency under extremely low-contrast or heavy depth noise that would better illustrate the limits of the one-way refinement. Furthermore, although stage-wise interaction evolution maps in B.7 show progressive focusing on salient regions no quantitative correlation is provided between the spatial concentration of activations at step T and the final per-image MAE or boundary F_beta to confirm that the Hamiltonian dynamics indeed drives the observed improvements. In addition, the paper acknowledges that generalization beyond RGB-D SOD with predefined roles was not tested, yet it would strengthen the claims to know whether any quick experiment was attempted on related tasks like RGB-T or even synthetic bimodal data to check if the spectral force and gated momentum transfer without major retuning. Finally, because the corruption robustness relies on auxiliary-only noise, it remains open whether the same graceful degradation holds when both modalities are simultaneously corrupted at moderate levels, which would be a more realistic sensor-failure scenario for practical deployment.

---

> ### Author Rebuttal · Authors · 2026-03-30
>
> We appreciate your feedback. We clarify: (i) SGC implementation and complex conjugation; (ii) HAF as controlled contraction rather than conservative simulation; (iii) bounded asymmetric feedback and role assignment; and (iv) matched-setting implementation, robustness, and efficiency.
>
> **W1: SGC implementation.**
> $G$ is initialized to ones, learned independently per HAF stage, and used without extra regularization. We did not observe degenerate collapse; removing learnable $G$ already hurts performance (Appendix B.2). The MHSA baseline is a matched replacement under the same architecture, pipeline, and hardware. To assess complex conjugation, we compare full SGC with a no-conjugate variant under default $T=4$:
>
> | Variant | DUT-RGBD ($S_m/F_\beta$) | NJU2K ($S_m/F_\beta$) |
> |---|---:|---:|
> | Full SGC | 0.953/0.963 | 0.938/0.945 |
> | SGC w/o complex conjugate | 0.950/0.961 | 0.935/0.942 |
>
> **W2: Leaky momentum refinement.**
> HAF is not a conservative Hamiltonian simulator, but a controlled refinement mechanism that realizes contraction under noisy auxiliary perturbations (Theorem 3.7, gain $\kappa$). Eqs. (10)–(11) favor stable multi-step refinement: the leaky term damps stale momentum, and the gate controls force injection. The gate is a two-layer MLP on $[q;p]$ with hidden dimension=head_dim, GELU, and element-wise sigmoid. Since $p$ is a block-local update state rather than a physical momentum with an external prior, we use neutral initialization $p^0=0$; $p$ is reset in each HAF block.
>
> | Init of $p^0$ | DUT-RGBD ($S_m/F_\beta$) | NJU2K ($S_m/F_\beta$) |
> |---|---:|---:|
> | Zero init | 0.953/0.963 | 0.938/0.945 |
> | Random init | 0.951/0.961 | 0.937/0.944 |
>
> **W3: Asymmetric residual coefficients.**
> $\alpha_{aux}$ and $\alpha_{main}$ are initialized as 0.8 and 0.2 and learned independently per stage. Under fixed $T=4$, the default asymmetric initialization is the most consistent overall; reversed asymmetry, symmetric initialization, or fixing $\alpha_{main}=0$ does not improve performance. This supports weak, bounded primary feedback rather than zero back-injection or strict symmetry. A full analysis of learned $\alpha$ distributions and Corollary 3.8 is beyond the current submission.
>
> For brevity, we report $S_m/F_\beta$.
>
> | Setting | DUT-RGBD | NJU2K |
> |---|---:|---:|
> | Default | 0.953/0.963 | 0.938/0.945 |
> | Reversed | 0.951/0.962 | 0.934/0.939 |
> | Symmetric | 0.952/0.963 | 0.935/0.942 |
> | Fixed $\alpha_{main}=0$ | 0.952/0.961 | 0.936/0.943 |
>
> **W4: Role assignment.**
> HAF targets asymmetric multimodal settings where a relatively stable modality guides a higher-variance one; in RGB-D SOD, this is depth-primary / RGB-auxiliary. The role-swap ablation is intentionally kept under a fixed protocol (same $\gamma$, gate design, and training setup) to isolate the effect of reversing the reliability ordering. The larger degradation on SIP than on NLPR is consistent with dataset characteristics, since SIP has finer structures and stronger distractions. For broader evidence beyond RGB-D SOD, see Reviewer LL3H (W1&W4) for the added RGB-T result under the same reliability-guided role assignment.
>
> **W5: Backbone and protocol.**
> Both RGB and depth branches use the same Swin backbone (embed_dim=128, depths=[2,2,18,2], num_heads=[4,8,16,32]), with HAF inserted at four stages (1024/512/256/128). Fixed $T=4$ is the main setting and best-$T$ is supplementary (Reviewer xV53, W1–W3). Plugin-style comparisons use no extra post-processing. All results are single-GPU and without AMP.
>
> **W6: Robustness protocol.**
> As clarified in Reviewer xV53 (W4), the robustness study uses Gaussian corruption on the auxiliary RGB stream only, injected before encoder feature extraction, while keeping depth clean; L0–L4 correspond to $\sigma=[0,5,10,20,30]$ in pixel space $[0,255]$. This is a controlled auxiliary-corruption stress test, not a full sensor-failure benchmark. Appendix B.8 already contains per-dataset degradation curves and AUC trends, which we will make more explicit in the revision. Significance tests versus methods such as SHIP would be a useful extension.
>
> **W7: Efficiency comparison.**
> Table 3 profiles the full CPNet pipeline under matched settings, not the fusion block alone. FLOPs use fvcore; latency/throughput/peak memory are measured in eval mode without AMP. The reported $B=2$ is the largest common batch size fitting the same GPU memory budget for both CDA and HAF, i.e., a matched common-budget comparison rather than HAF’s limit. Larger-batch $T$-scaling is reported in Appendix B.6.
>
> **W8: Qualitative analysis.**
> Fig. 9/B.9 are illustrative rather than a systematic failure analysis; we will add representative failures and clarify this scope. Appendix B.7 is qualitative support for primary-guided refinement, not quantitative proof with respect to per-image MAE or boundary $F_\beta$. See also Reviewer LL3H (W1&W4) for the added RGB-T result and Reviewer ZRao (W1) for the added both-modality corruption analysis.

---

> > ### Author Rebuttal · Reviewer_Drqf · 2026-04-04
> >
> > na

---

> > > ### Author Response · Authors · 2026-04-04
> > >
> > > Thank you very much for your follow-up and for marking the concerns as fully resolved. We sincerely appreciate your acknowledgement that our rebuttal has adequately addressed your questions. If there are any remaining concerns, please feel free to let us know, and we would be glad to provide further clarification. We would also sincerely appreciate it if you could reconsider the score in light of the clarifications provided in our rebuttal.

---

### Decision · Program_Chairs · 2026-04-30

**Decision:**

Accept (regular)

**Comment:**

This paper presented an asymmetric fusion to handle the modality imbalance issue —when an auxiliary stream is substantially noisier than a designated primary stream. Experiments on RGB-D saliency detection tasks demonstrated the effectiveness of the proposed approach.

Three reviewers were satisfied with the rebuttal. One reviewer had a question: the experiment was only limited to RGB–D salient object detection. The further feedback from the authors provided more results.

As the proposed approach was proposed for saliency detection, it would be better to explicitly mention it in the title or more in the abstract.